# RNA-Protein Interactome at the Hepatitis E Virus Internal Ribosome Entry Site

Shiv Kumar,[a] Rohit Verma,[a] Sandhini Saha,[b] Ashish Kumar Agrahari,[c] Shivangi Shukla,[a] Oinam Ningthemmani Singh,[a] Umang Berry,[a] Anurag,[a] Tushar Kanti Maiti,[b] Shailendra Asthana,[c] C. T. Ranjith-Kumar,[d] (ORCID) Milan Surjit[a]

[a]Virology Laboratory, Translational Health Science and Technology Institute, NCR Biotech Science Cluster, Faridabad, India
[b]Laboratory of Functional Proteomics, Regional Centre for Biotechnology, NCR Biotech Science Cluster, Faridabad, India
[c]Noncommunicable Disease Group, Translational Health Science and Technology Institute, NCR Biotech Science Cluster, Faridabad, India
[d]University School of Biotechnology, Guru Gobind Singh Indraprastha University, New Delhi, India

**ABSTRACT** Multiple processes exist in a cell to ensure continuous production of essential proteins either through cap-dependent or cap-independent translation processes. Viruses depend on the host translation machinery for viral protein synthesis. Therefore, viruses have evolved clever strategies to use the host translation machinery. Earlier studies have shown that genotype 1 hepatitis E virus (g1-HEV) uses both cap-dependent and cap-independent translation machineries for its translation and proliferation. Cap-independent translation in g1-HEV is driven by an 87-nucleotide-long RNA element that acts as a noncanonical, internal ribosome entry site-like (IRESl) element. Here, we have identified the RNA-protein interactome of the HEV IRESl element and characterized the functional significance of some of its components. Our study identifies the association of HEV IRESl with several host ribosomal proteins, demonstrates indispensable roles of ribosomal protein RPL5 and DHX9 (RNA helicase A) in mediating HEV IRESl activity, and establishes the latter as a bona fide internal translation initiation site.

**IMPORTANCE** Protein synthesis is a fundamental process for survival and proliferation of all living organisms. The majority of cellular proteins are produced through cap-dependent translation. Cells also use a variety of cap-independent translation processes to synthesize essential proteins during stress. Viruses depend on the host cell translation machinery to synthesize their own proteins. Hepatitis E virus (HEV) is a major cause of hepatitis worldwide and has a capped positive-strand RNA genome. Viral nonstructural and structural proteins are synthesized through a cap-dependent translation process. An earlier study from our laboratory reported the presence of a fourth open reading frame (ORF) in genotype 1 HEV, which produces the ORF4 protein using a cap-independent internal ribosome entry site-like (IRESl) element. In the current study, we identified the host proteins that associate with the HEV-IRESl RNA and generated the RNA-protein interactome. Through a variety of experimental approaches, our data prove that HEV-IRESl is a bona fide internal translation initiation site.

**KEYWORDS** hepatitis E virus, IRES-mediated translation, RNA-protein interaction

Protein synthesis is an essential process for survival and growth of all organisms, and viruses depend on the host cell machinery for protein synthesis. Proteins are synthesized by the process of translation, which involves four distinct steps, including initiation, elongation, termination, and recycling of the ribosomes. In eukaryotes under normal conditions, the majority of mRNAs are translated by recognition of the m⁷G $(5')_{PPP}(5')N$ (cap) structure present at the 5' end of the mRNA by the process of cap-dependent translation. Many cellular and viral mRNAs are translated by cap-independent mechanisms by using distinct *cis*-acting regulatory elements, such as cap-independent

Address correspondence to Milan Surjit, milan@thsti.res.in.

The authors declare no conflict of interest.

translational enhancers (CITEs), $N^6$-methyladenine (m$^6$A) modification of the mRNAs, or internal ribosome entry sites (IRESs). Initiation of cap-dependent translation occurs through a ribosomal scanning mechanism in which the 48S initiation complex moves along the mRNA from the m$^7$G cap in the 5′ to 3′ direction to locate a suitable AUG initiation codon. Following AUG recognition, through a series of steps, the 80S ribosome complex is assembled on the mRNA with initiator methionyl-tRNA (met-tRNAi) bound to the AUG codon at the P-site, leading to translation elongation (1). The CITEs are located in either 5′ or 3′ untranslated regions (UTRs) of mRNAs, and they initiate cap-independent translation by recruiting initiation factors to the uncapped mRNAs, followed by 5′-end ribosomal scanning-mediated identification of the initiation codon. Under conditions of stress, the cap-independent translation of many cellular mRNAs, such as *c-myc*, *Apaf1*, and *XIAP*, seems to occur through the above mechanism (2, 3). A variant of the eukaryotic initiation factor 4G (eIF4G) has been shown to be important for cap-independent translation of *c-myc* (4). Recently, yeast eIF4G1 was shown to mediate cap-independent translation from the black beetle virus 5′-UTR (5). Translation of 5′- or 3′-UTR m$^6$A-modified mRNAs also occurs in a cap-independent manner (6–8).

Cap-independent, IRES-mediated translation has been reported in many positive-strand RNA viruses, which contain an uncapped genomic RNA. The 5′-UTR in the genomic RNA of these viruses contains highly structured RNA elements, which directly recruit initiation factors and promote translation through a scanning-independent process, with the exception of the type I IRES, which depends on the ribosomal scanning process. IRESs are divided into five major types based on their mode of ribosome recruitment and RNA structure. The type I and type II IRESs are found in picornaviruses, such as polio virus (PV) and foot and mouth disease virus (FMDV), respectively. The PV IRES harbors six stem-loops (SLs) designated domains I to VI. Domain I forms a unique clover leaf structure and is critical for replication of both positive- and negative-sense RNA. Domains II to VI are responsible for PV IRES activity. During PV infection, viral 2A$^{pro}$ cleaves the eIF4E binding N-terminal domain of eIF4G without affecting its eIF3/eIF4A binding property. Stable association of eIF4G with PV IRES domain V promotes binding of other initiation factors, leading to formation of the 43S preinitiation complex. The FMDV IRES is a well-studied example of a type II IRES. Domain IV of the FMDV IRES binds the scaffold protein eIF4G. The 3C$^{pro}$ and L$^{pro}$ of FMDV cleave eIF4G similar to 2A$^{pro}$ of type I IRESs. Notably, the FMDV IRES skips ribosomal scanning; instead, the IRES-proximal stem-loop formation brings 84 nucleotides (nt) downstream of AUG close to the first AUG to start translation by direct ribosome transfer. Other identified *cis*-acting elements for type II IRES activity are the GNRA, RAAA, and C-rich loops in domain III (9). The type III IRES is present in the 5′ UTR of the hepatitis A virus genome (10) and requires eIF4E binding for translation initiation (11). Type IV IRESs have been reported in members of the *Flaviviridae* family, classified as the hepatitis C virus (HCV)/HCV-like IRES. The 5′ UTR of HCV contains four domains; domains I and II are involved in viral replication, while domains III and IV are involved in IRES activity (12, 13). The HCV/HCV-like IRESs are shorter than type I and II IRESs. Domains II and III contain several subdomains for interaction with the 40S ribosomal subunit. Type V IRESs include the long intergenic region (IGR) IRES, which is found between two open reading frames in viral genomes and is conserved in the *Dicistroviridae* family (12, 13). IGR IRESs are the smallest IRES elements (∼180 nt long) and consist of three pseudoknots. IGR IRESs directly bind to ribosomes and initiate translation with alanine-tRNAi (ala-tRNAi) instead of met-tRNAi, without involving the eIFs (14, 15).

Many viruses have evolved the ability to use both cap-dependent and cap-independent translation processes depending on the state of the cell. Notably, although dengue virus and Zika virus genomic RNAs are usually translated through a cap-dependent mechanism, their 5′ UTRs also contain IRES elements, which maintain viral translation under unfavorable conditions where cap-dependent translation is inhibited by the host (16–18). Similarly, genomic RNA of human immunodeficiency virus (HIV) type 1 and simian immunodeficiency virus (SIV) contains IRES elements in their 5′ leader sequences,

which maintain viral translation when cap-dependent translation is inhibited (19–22). In addition, recent studies have identified functional IRES elements within open reading frames of viral protein-coding genes. For example, five functional IRES elements are located within the coding region of nonstructural and structural proteins of human rhinovirus 16, which are active under conditions of endoplasmic reticulum (ER) stress (23, 24). The human cytomegalovirus pUL138 protein is translated by an IRES under conditions of stress (25). Further, coat protein of the turnip crinkle virus (TCV) is translated through an unstructured IRES (26).

Hepatitis E virus (HEV) is a positive-strand RNA virus of the family *Hepeviridae*. The viral genome is capped at the 5′ end followed by a short UTR of 25 nucleotides. The viral nonstructural proteins are synthesized from open reading frame 1 (ORF1) located between nucleotides 26 and 5107. Viral capsid protein is produced by ORF2 located at the 3′ end between nucleotides 5145 and 7127. A short 3′ UTR of 65 nucleotides is present downstream of ORF2, followed by a poly(A) tail (27). A third ORF is present between ORF1 and ORF2, which produces ORF3 protein, an accessory protein that associates with the host tumor susceptibility gene 101 and facilitates the release of progeny virus (28–30). The above genome organization is conserved among the 7 genotypes of HEV. In contrast to the zoonotic nature of some HEV genotypes, genotype 1 HEV (g1-HEV) infection is restricted to humans, and g1-HEV does not replicate efficiently in mammalian cell lines (31).

Our earlier study demonstrated the coexistence of both cap-dependent and cap-independent translation in g1-HEV (32). Although HEV nonstructural and structural proteins are synthesized by a cap-dependent translation process, a protein essential for g1-HEV replication (ORF4 protein) is synthesized from an overlapping ORF located within viral ORF1 through a cap-independent translation process. An 87-nucleotide conserved RNA regulatory element located upstream of the ORF4 coding region was found to drive cap-independent translation of the ORF4 protein. Interestingly, although ORF4 protein production in g1-HEV-infected cells was markedly enhanced after treatment of cells with ER stress-inducing compounds, HEV IRES-like (IRESl) functioned efficiently irrespective of ER stress inducer treatment in *in vitro* translation assays or bicistronic reporter-based assays. Analysis of cap-independent translation driven by the HEV RNA regulatory element by bicistronic reporter assays along with site-directed mutagenesis-mediated mapping of the regulatory RNA sequence suggested that the HEV RNA regulatory element is an IRES element. However, considering the small size of the regulatory element and lack of strong homology with canonical IRES elements, it was designated an IRESl element. The current study was designed to further explore the mechanism of HEV IRESl activity and verify its function as a bona fide internal translation initiation site. Host interaction partners of HEV IRESl were identified, and their importance in IRESl-mediated translation was evaluated. Subsequent experiments showed the association of HEV IRESl RNA with actively translating ribosomes, supporting its role as a translation initiation element. The significance of these findings in validating the role of HEV IRESl as a bona fide translation initiation site is discussed.

## RESULTS

**Identification of host proteins that interact with the HEV IRESl element.** Two independent experimental approaches (an RNA-protein interaction detection [RaPID] assay and a yeast three-hybrid assay [Y3H]) were used to identify host proteins that interact with HEV IRESl RNA. The RaPID assay involves biotinylation of IRES-interacting proteins, followed by their identification by liquid chromatography with tandem mass spectroscopy (LC-MS/MS) (33). The RaPID assay was optimized in our laboratory (34). Eighty-seven nucleotides of the g1-HEV genome (GenBank ID: AF444002.1) corresponding to the IRESl region were cloned into the pRMB vector between the BirA ligase-binding stem-loop (RMB-SLI and RMB-SLII) sequences (Fig. 1A and C). Analysis of the secondary structure of the fusion RNA sequence using "mFold" indicated that HEV IRESl and the BirA ligase-binding stem-loops retained their distinct folding characteristics (Fig. 1A). An unrelated sequence in the g1-

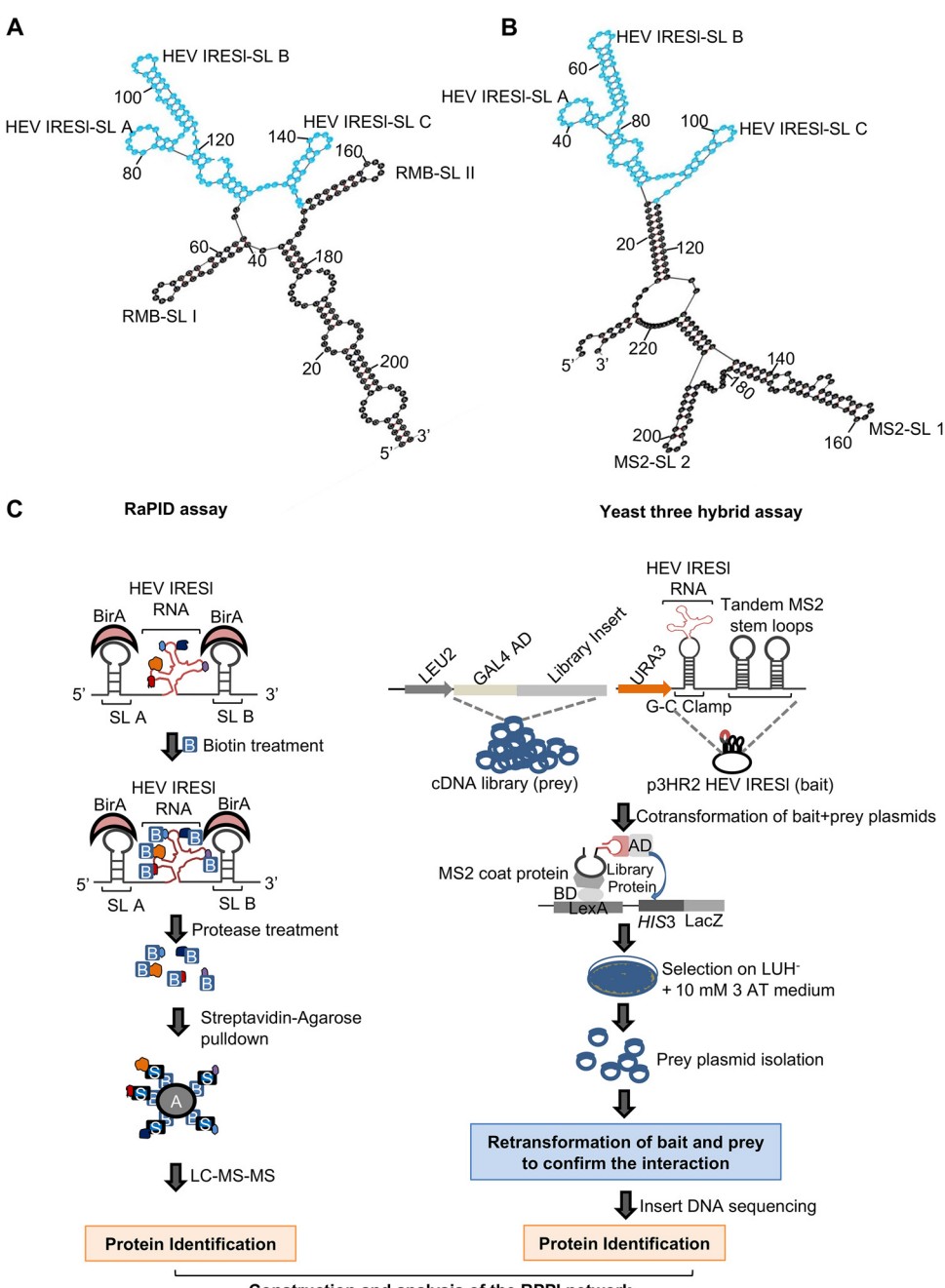

**FIG 1** Identification of host interaction partners of the HEV IRESl element by RaPID and yeast three-hybrid assays. (A) Schematic of the predicted secondary structure of HEV IRESl RNA (highlighted in blue) fused to the BirA-binding RNA motifs (RMB-SL1 and RMB-SL2). Stem-loops of HEV IRESl are denoted as SL A, SL B, and SL C. (B) Schematic of the predicted secondary structure of the HEV IRESl RNA (highlighted in blue) fused to the MS2 coat protein-binding RNA motifs (MS2-SL1 and MS2-SL2). Stem-loops of the HEV IRESl are denoted as SL A, SL B, and SL C. (C) Workflow to identify the HEV IRESl-binding host proteins; B, biotin; A, agarose; S, streptavidin; AD, GAL4 activation domain; BD, GAL4 DNA-binding domain; L, leucine; U, uracil; H, histidine; 3-AT, 3-amino-1,2,4-triazole; −, deficiency in the medium; +, supplemented in the medium.

HEV genome (618 to 738 nucleotides from the 5′ end, designated "control RNA") was also cloned into the pRMB vector to be used as a control to monitor specificity of the assay, and biotin-untreated cells were used as an additional control. Experiments were performed two times, and samples of each experiment were run in triplicate.

The quality of the MS data between replicates of different samples was checked by Pearson correlation analysis, which showed good correlation (average range of 0.2 to

1.0). Proteins with at least one biotinylated peptide with a posterior error probability (PEP) score greater than or equal to the median value in the Gaussian smoothing curve of each sample were selected for further analysis. Specific interaction partners of HEV IRESl RNA were selected by subtracting the biotinylated HEV IRESl data set from the biotinylated HEV (618 to 738), unbiotinylated HEV (618 to 738), and unbiotinylated HEV IRESl data sets (Fig. 2A; Fig. S1 and Table S1 in the supplemental material). The obtained HEV IRESl RNA-binding protein data set was further analyzed to select only those proteins that were represented by one or more unique peptides and a "Prot score" of 30 or more. Thus, 43 unique proteins were identified as interaction partners of HEV IRESl RNA (Fig. 2B; Table 1).

Because the RaPID assay may have the limitation of not identifying all proteins buried inside the RNA-protein complex, a Y3H assay was used as an alternate approach to unbiasedly identify the direct interaction partners of HEV IRESl RNA. A human liver cDNA library was screened using the Y3H assay (Fig. 1C) (35, 36). In the Y3H assay, the bait RNA is flanked by two copies of the MS2 coat protein-binding RNA elements. Analysis of the RNA sequence containing the fusion of the HEV-IRESl RNA and the MS2 coat protein-binding stem-loop RNA sequences using "mFold" indicated that their secondary structure profiles remain unaltered in the fusion RNA (Fig. 1B). Self-activation of the *lacZ* and *HIS3* reporter genes by HEV IRESl RNA was checked by cotransforming the plasmids encoding HEV IRESl (p3HR2-HEV IRESl) and the GAL4 activation domain (pACT2) into chemically competent YBZ1 yeast cells, followed by selection of the cotransformants on medium lacking leucine, uracil, and histidine (LUH⁻) and assessment of $\beta$-galactosidase activity. No growth of colonies was seen on LUH⁻ plates nor was $\beta$-galactosidase activity observed, indicating that there was no self-activation of the reporter genes by the bait RNA (Fig. 2C). Next, the bait RNA-expressing YBZ1 cells were transformed with the human liver cDNA library. A total of $4 \times 10^6$ clones were screened, and cotransformants were selected on LUH⁻ medium supplemented with 10 mM 3-amino-1,2,4 triazole (3-AT). Note that 3-AT is a competitive inhibitor of the histidine biosynthesis pathway. Therefore, addition of 3-AT allows growth of the cotransformants that show strong interaction between the bait RNA and the prey protein. Three hundred and ninety-five colonies were obtained in the primary screening, of which the human liver cDNA insert could be detected in 285 colonies (Table S2). Restriction pattern analysis of the prey plasmid DNA isolated from the 285 colonies showed that the unique cDNA insert was present in 75 plasmids. Analysis of the cDNA insert sequences identified 8 protein-coding genes, which were in frame with the GAL4 activation domain (Table S2). Retransformation of those 8 plasmids along with the p3HR2-HEV IRESl plasmid showed that all of them were able to interact with the HEV IRESl (Fig. 2C). The eight proteins include six ribosomal proteins (RPL5, RPL26, RPL41, RPS3A, RPS7, and RPS15A), peptidylprolyl isomerase G (PPIG), and cyclic-GMP response element binding protein (GREBP) (Table 1).

Further, specificity of the interaction of eight Y3H-identified proteins with the HEV IRESl was evaluated in two ways: (i) by measuring the interaction of those proteins with an RNA sequence complementary to the HEV IRESl (HEV IRESl⁻) and (ii) by measuring the interaction of those proteins with the known IRES sequences from the genotype 3A hepatitis C virus (HCV) and the foot and mouth disease virus (FMDV). All sequences were cloned into the p3HR2 vector upstream of the MS2 coat protein-binding stem-loop RNA sequences (Fig. S2). No significant interaction was observed between HEV IRESl⁻ RNA and the host proteins (Fig. 2C). Y3H analysis revealed that only RPS3A interacts with equal strength with all three RNA sequences (HEV IRESl, FMDV IRES, and HCV IRES) (Fig. 3; Table S3). RPL41 and RPS15A interact exclusively with HEV IRESl RNA, while RPL5 and RPS7 were found to interact strongly with HEV IRESl RNA and weakly with the HCV IRES. RPL26 interacts moderately and strongly with the HEV IRESl and the HCV IRES, respectively, whereas it did not interact with the FMDV IRES. GREBP associates strongly with the HEV IRESl and weakly with both FMDV IRES and HEV IRESl. PPIG also showed a difference in the strength of interaction among the three RNA sequences in the following order: HCV IRES (strong interaction), HEV IRESl (weak interaction), and FMDV IRES (no

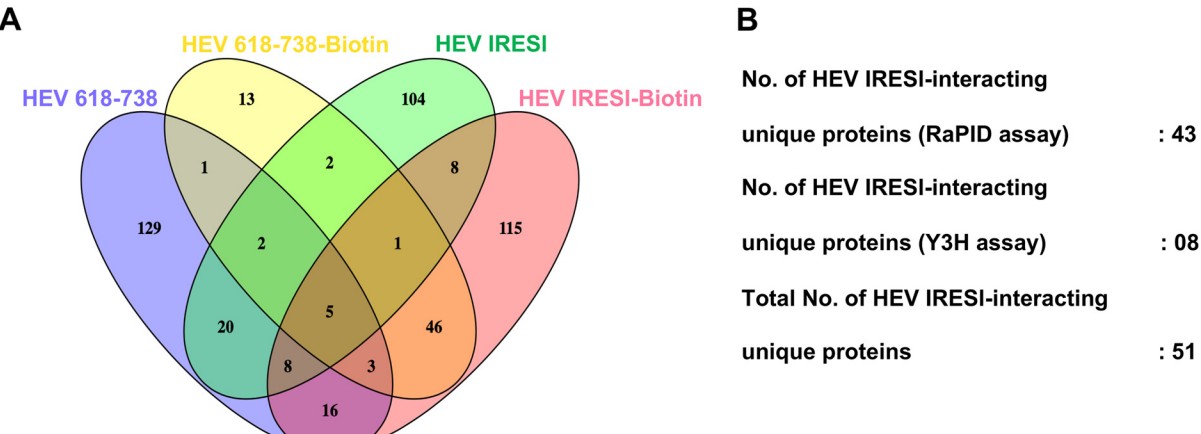

**A**

**B**

No. of HEV IRESI-interacting

unique proteins (RaPID assay)          : 43

No. of HEV IRESI-interacting

unique proteins (Y3H assay)            : 08

Total No. of HEV IRESI-interacting

unique proteins                        : 51

**C**

| Yeast Cotransformants/ Selection Media | LU⁻ | LUH⁻ + 3AT (mM) | | | Relative β-galactosidase units (mean ± SEM) |
|---|---|---|---|---|---|
| | | 0 | 10 | 20 | 10 30 50 70 90 110 130 150 |
| p3HR2- HEV IRESI + pACT2 | | | | | |
| p3HR2- HEV IRESI⁻ + pACT2 | | | | | |
| p3HR2 + pACT2 | | | | | |
| p3HR2-HEV IRESI + pACT2-GREBP | | | | | |
| p3HR2-HEV IRESI⁻ + pACT2 GREBP | | | | | P=0.025 |
| p3HR2 + pACT2-GREBP | | | | | |
| p3HR2-HEV IRESI + pACT2-PPIG | | | | | |
| p3HR2-HEV IRESI⁻ + pACT2-PPIG | | | | | P=0.02 |
| p3HR2 + pACT2-PPIG | | | | | |
| p3HR2-HEV IRESI + pACT2-RPL5 | | | | | |
| p3HR2-HEV IRESI⁻ + pACT2-RPL5 | | | | | P=0.015 |
| p3HR2 + pACT2-RPL5 | | | | | |
| p3HR2-HEV IRESI + pACT2-RPL26 | | | | | |
| p3HR2-HEV IRESI⁻ + pACT2-RPL26 | | | | | P=0.002 |
| p3HR2 + pACT2-RPL26 | | | | | |
| p3HR2-HEV IRESI + pACT2-RPL41 | | | | | |
| p3HR2-HEV IRESI⁻ + pACT2-RPL41 | | | | | P=0.006 |
| p3HR2 + pACT2-RPL41 | | | | | |
| p3HR2-HEV IRESI + pACT2 RPS3A | | | | | |
| p3HR2-HEV IRESI⁻ + pACT2-RPS3A | | | | | P=0.001 |
| p3HR2 + pACT2-RPS3A | | | | | |
| p3HR2-HEV IRESI + pACT2-RPS7 | | | | | |
| p3HR2-HEV IRESI⁻ + pACT2-RPS7 | | | | | P=0.002 |
| p3HR2 + pACT2-RPS7 | | | | | |
| p3HR2-HEV IRESI + pACT2 RPS15A | | | | | |
| p3HR2-HEV IRESI⁻+ pACT2 RPS15A | | | | | P=0.0009 |
| p3HR2 + pACT2-RPS15A | | | | | |

**FIG 2** Identification of HEV IRESI RNA-binding proteins by RaPID and Y3H assays. (A) Venny analysis of the HEV IRESI RNA-binding proteins identified by the RaPID assay. Blue- and green-colored ovals represent the numbers of host proteins identified by RaPID assay using HEV 618-738 RNA (blue)

interaction) (Fig. 3; Table S3). Note that growth of colonies on LUH$^-$ + 3-AT medium was considered for evaluating the strength of interaction, as there were variations in $\beta$-galactosidase activity between different interaction partners.

To further ascertain that HEV IRESl interaction partners identified by RaPID and Y3H are bona fide candidates, some of the interactions were further validated by an *in vitro* pulldown assay using *in vitro*-transcribed biotinylated HEV IRESl RNA as bait. Nonbiotinylated HEV IRESl and biotinylated control RNA (nucleotides 618 to 738 in the g1-HEV genome) as well as cell lysate only (mock) were used as controls to ensure specificity of the assay (Fig. 4A and B). Further, glyceraldehyde-3-phosphate dehydrogenase (GAPDH) protein, which does not interact with HEV IRESl RNA, was used as a negative control to test specificity of the pulldown assay. GAPDH protein was not detected in the pulldown sample, supporting specificity of the pulldown assay (Fig. 4C). All 10 HEV IRESl RNA-interacting proteins tested by the pulldown assay showed interaction with the HEV IRESl RNA (Fig. 4C). Seven of these interactions were originally identified by Y3H assay (RPS3A, RPS7, RPL26, RPL41, PPIG, GREBP, and RPL5), and three were identified by RaPID assay (RPL24, LARP4, and DHX9) (Fig. 4C). These data further support that the proteins identified by RaPID and Y3H assays are bona fide interaction partners of HEV IRESl RNA.

**Construction of the HEV IRESl RNA-host protein interaction network and analysis of the interactome.** Forty-three and 8 host proteins identified as interaction partners of the HEV-IRESl RNA by RaPID and Y3H assay, respectively, were clubbed and imported to Cytoscape to generate the RNA-protein-protein interaction (RPPI) network, as described earlier (29). Analysis of the different network characteristics revealed that the HEV IRESl RPPI network contained 51 nodes and 74 edges, with an average node degree of 3.36. The average clustering coefficient was 0.44, and the protein-protein interaction (PPI) enrichment $P$ value was $1.11 \times 10^{-16}$ (Fig. 4D). The above network parameters suggest that the HEV IRESl RPPI network is highly connected and suitable for further downstream studies.

Next, Gene ontology (GO) and Reactome pathway analysis of the HEV IRESl RPPI network was performed using the "Enrichr" tool to determine the significantly enriched processes/pathways. Proteins involved in translation and mRNA catabolic processes were enriched in the GO biological processes category, supporting involvement of the HEV IRESl element in protein synthesis (Fig. 4E; the top 10 biological processes are shown). Similarly, the top Reactome pathways include translation, eukaryotic translation elongation, and selenoamino acid metabolism (Fig. 4F; the top 10 Reactome pathways are shown). Analysis of the data set using the gene set enrichment analysis (GSEA) tool also showed translation and peptide metabolic process as top hits, in agreement with the Enrichr output (Fig. S3A and B). Components of both small and large subunits of the ribosome, such as RPS7, RPS3A, RPS15A and RPL5, RPL24, RPL26, and RPL41, interact with HEV IRESl. Components of the tRNA synthetase complex, such as threonyl-tRNA synthetase (TARS), valyl-tRNA synthetase (VARS), aminoacyl tRNA synthetase complex-interacting multifunctional protein 1 (AIMP1), and a component of the translation elongation complex (eukaryotic elongation factor 2 [EEF2]), also interact with the HEV IRESl, further supporting its involvement in protein synthesis (Table 1). In addition, a literature search of IRES-binding host proteins revealed that four proteins identified in our study are known to bind other IRESs. DHX9 and RPA1 bind to the IRES of HCV and FMDV, RPS3A associates with the HCV IRES, and RPL26 interacts with the p53 IRES (Table 1, highlighted in bold) (37–39).

**FIG 2** Legend (Continued)
and HEV IRESl RNA (green) in the absence of biotin treatment. HEV 618-738-biotin (yellow) and HEV-IRESl-biotin (pink) represent the number of host proteins identified by RaPID assay using HEV 618 to 738 RNA and HEV IRESl RNA in biotin-treated cells. Proteins overlapping between different data sets are shown in respective merged color, and unique proteins identified in each data set are shown in single color. (B) Summary of HEV IRESl RNA-binding proteins identified by the RaPID and Y3H assays. (C) Y3H assay-mediated confirmation of the interactions between HEV IRESl RNA and host proteins identified by screening of the human liver cDNA library. The YBZ1 strain was transformed in the indicated combinations and plated on medium lacking leucine and uracil (LU$^-$). Four random colonies from each cotransformant plate were replica plated onto medium lacking leucine, uracil, and histidine (LUH$^-$) and supplemented with 5 mM or 10 mM 3-amino-1,2,4-triazole (3-AT). The same colonies were used in liquid $\beta$-galactosidase assays. Relative $\beta$-galactosidase units are plotted as mean $\pm$ SEM.

**TABLE 1** Host interaction partners of the HEV IRESI element

| Gene name | Description | Prot score | No. of biotinylated peptides | No. of unique biotinylated peptides |
|---|---|---|---|---|
| HEV IRESI-interacting host proteins identified by the RaPID assay[a] | | | | |
| EEF2 | Elongation factor 2 | 1,568 | 8 | 6 |
| TUBA3C | Tubulin alpha-3C/D chain | 929 | 1 | 1 |
| HNRNPA2B1 | Heterogeneous nuclear ribonucleoproteins A2/B1 | 338 | 2 | 2 |
| **DHX9** | ATP-dependent RNA helicase A | 285 | 8 | 7 |
| TARS | Threonyl-tRNA synthetase, cytoplasmic | 285 | 7 | 5 |
| **RPL7A** | 60S ribosomal protein L7a | 262 | 4 | 3 |
| BAT2L2 | Protein BAT2-like 2 | 75 | 1 | 1 |
| NUFIP2 | Nuclear fragile X mental retardation-interacting protein 2 | 75 | 2 | 1 |
| VARS | Valyl-tRNA synthetase | 56 | 12 | 8 |
| ARPP19 | cAMP-regulated phosphoprotein 19 | 53 | 1 | 1 |
| ENSA | Alpha-endosulfine | 53 | 1 | 1 |
| RPL24 | 60S ribosomal protein L24 | 53 | 1 | 1 |
| TARSL2 | Probable threonyl-tRNA synthetase 2, cytoplasmic | 53 | 5 | 5 |
| YTHDF2 | YTH domain family protein 2 | 53 | 3 | 2 |
| HSPH1 | Heat shock protein 105 kDa | 50 | 9 | 6 |
| CAT | Catalase | 49 | 2 | 2 |
| DDX10 | Probable ATP-dependent RNA helicase DDX10 | 42 | 15 | 9 |
| LARP4 | La-related protein 4 | 42 | 6 | 1 |
| NDUFAF2 | Mimitin, mitochondrial | 41 | 1 | 1 |
| DBNL | Drebrin-like protein | 39 | 3 | 1 |
| LBR | Lamin-B receptor | 38 | 1 | 1 |
| NSF | Vesicle-fusing ATPase | 37 | 1 | 1 |
| CEACAM16 | Carcinoembryonic antigen-related cell adhesion molecule 16 | 35 | 1 | 1 |
| VWA5B1 | von Willebrand factor A domain-containing protein 5B1 | 35 | 5 | 5 |
| CCDC65 | Coiled-coil domain-containing protein 65 | 34 | 5 | 5 |
| PUM1 | Pumilio homolog 1 | 34 | 4 | 2 |
| ARSJ | Arylsulfatase J | 33 | 2 | 1 |
| **RPA1** | Replication protein A 70 kDa DNA-binding subunit | 33 | 4 | 4 |
| TXLNG | Gamma-taxilin | 33 | 17 | 10 |
| AIMP1 | Aminoacyl tRNA synthetase complex-interacting multifunctional protein 1 | 32 | 5 | 3 |
| BIVM | Basic immunoglobulin-like variable motif-containing protein | 32 | 7 | 3 |
| CCDC67 | Coiled-coil domain-containing protein 67 | 32 | 8 | 8 |
| RSBN1L | Round spermatid basic protein 1-like protein | 32 | 1 | 1 |
| SH3PXD2A | SH3 and PX domain-containing protein 2A | 32 | 1 | 1 |
| UBAP2 | Ubiquitin-associated protein 2 | 32 | 2 | 1 |
| GLS | Glutaminase kidney isoform, mitochondrial | 31 | 6 | 1 |
| MRE11A | Double-strand break repair protein MRE11A | 31 | 1 | 1 |
| ARL6IP5 | PRA1 family protein 3 | 31 | 2 | 1 |
| CENPP | Centromere protein P | 30 | 2 | 1 |
| ERP29 | Endoplasmic reticulum resident protein 29 | 30 | 2 | 1 |
| LASP1 | LIM and SH3 domain protein 1 | 30 | 1 | 1 |
| TMEM59L | Transmembrane protein 59-like | 30 | 2 | 1 |
| HEV IRESI-interacting host proteins identified by the Y3H assay | | | | |
| GREBP | cGMP-response element-binding protein | | | |
| PPIG | Peptidylprolyl isomerase G | | | |
| RPL5 | Ribosomal protein L5 | | | |
| **RPL26** | Ribosomal protein L26 | | | |
| RPL41 | Ribosomal protein L41 | | | |
| **RPS3A** | Ribosomal protein S3A | | | |
| RPS7 | Ribosomal protein S7 | | | |
| RPS15A | Ribosomal protein S15A | | | |

[a]RaPID data have been sorted on the basis of Prot score. Bold font represents proteins known to interact with IRES elements.

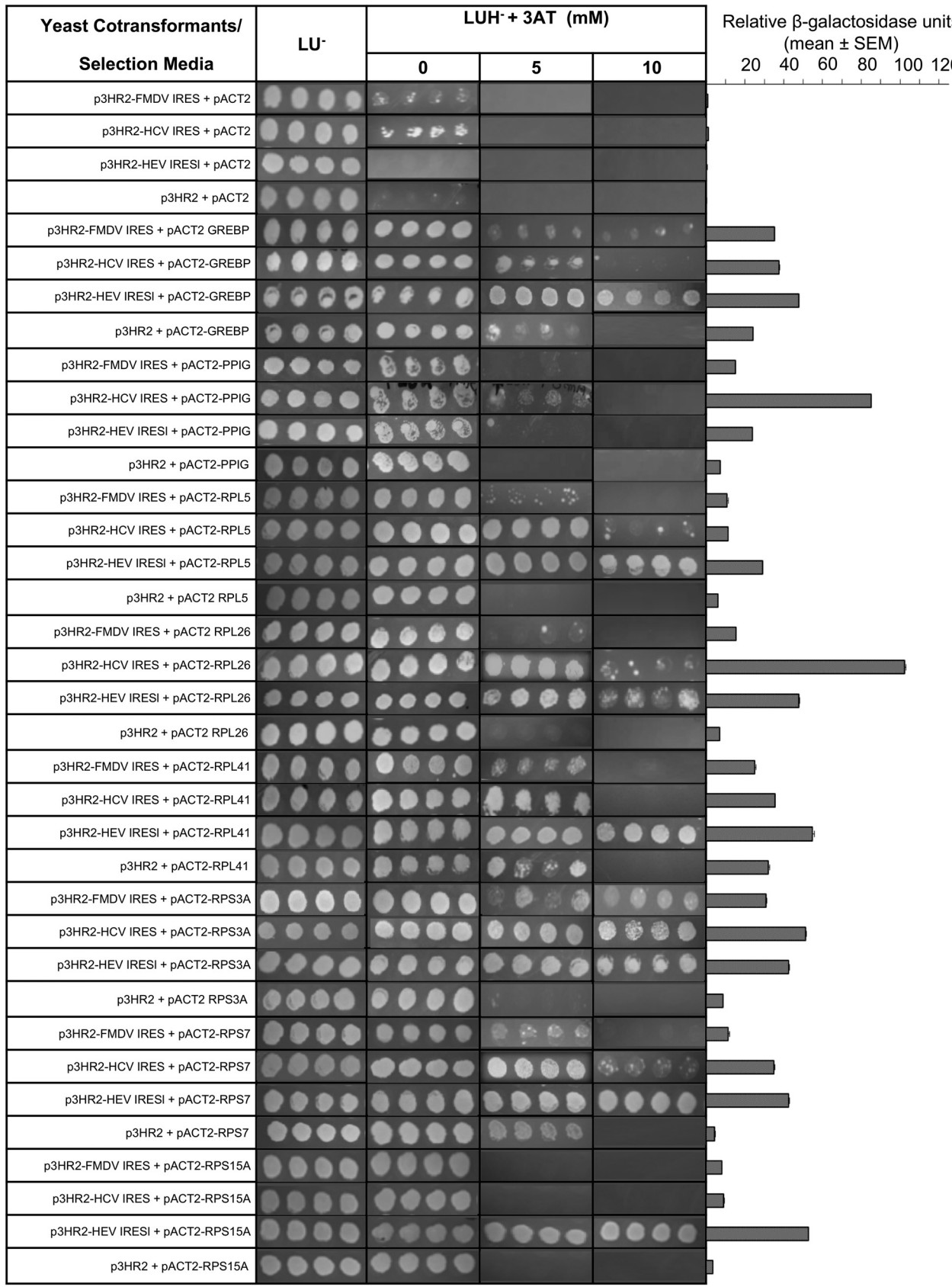

**FIG 3** Assessment of the ability of the HEV IRESI-binding host proteins (identified by the human liver cDNA library screening) to interact with the HCV IRES and FMDV IRES. The YBZ1 strain was transformed in the indicated combinations and plated on medium lacking leucine and uracil

**Interrogation of the interaction between HEV IRESl RNA and host ribosomal proteins by *in silico* analyses.** *In silico* molecular docking has been used to determine the complex structures of RNA-protein interactions (40). We used HDOCK, a web-based tool that is a hybrid docking algorithm of template-based modeling, to determine the interaction between HEV IRESl RNA and four of the identified proteins (RPL5, RPL26, RPS3A, and RPS7) (41). HEV IRESl showed direct interaction with RPL5, RPL26, RPS3A, and RPS7 proteins (Fig. 5A to D). All four complexes were analyzed for identification of the hot spot residues and were compared based on interaction mapping and binding energy analysis. All four complexes were quantified in terms of their interaction by mapping the critical polar contacts. The key nucleotides of each complex were mapped, and, notably, RPS3A, RPL26, and RPL5 were found to interact with HEV IRESl through a combination of common and unique nucleotides. Nucleotides U60, U29, G30, and U59 were common in at least two complexes. G50 and C44 are the unique nucleotides that contribute significantly in RPL5; U22, G51, and C68 are the unique nucleotides that contribute significantly in RPS3A; A38, C41, and U66 are the unique nucleotides that contribute significantly in RPL26; and U9, C10, G12, G53, and G56 are the unique nucleotides that contribute significantly in RPS7 (Fig. 5F to I). We also quantified the overall contributions by measuring the surface contacts between the HEV IRESl nucleotides and proteins (Fig. S3C). Here, the main focus was to identify the common nucleotides that were involved in contacts with the amino acids of the different ribosomal proteins. Interestingly, we found that nucleotides A8, U9, C28, G51, G52, U59, U60, G61, U67, C68, and C69 were common in at least three proteins. Nucleotides U29, G30, G31, A38, C41, A42, G50, C55, A57, U62, and A85 were common among two proteins (Fig. S3C). Among the identified nucleotides, some were engaged with two or more amino acids, such as C10, U22, U29, G30, U59, and U60, indicating their importance in mediating the interaction of HEV IRESl RNA with the respective ribosomal protein (Fig. 5F to I). Physicochemical analysis of the interacting amino acids revealed that most of the interacting residues were either basic (K/R) or aromatic (Y/F/H) in nature, and each complex involves at least one serine residue. To ensure specificity of the *in silico* analysis, GAPDH was used as a control protein to dock with HEV IRESl RNA. GAPDH is known to bind to a variety of RNAs, including the 5′ nontranslated region (NTR) of hepatitis A virus (42). GAPDH showed weak interaction with HEV IRESl RNA, involving nucleotides G26, C27, G45, and U76, which were distinct from those nucleotides mediating the interaction of HEV IRESl RNA with the ribosomal proteins (Fig. 5E and J). Further investigation of the binding energy of each complex showed that the order of binding affinity between the five complexes was RPL5 (−391.48 kcal/mol), RPL26 (−363.43 kcal/mol), RPS3A (−341.19 kcal/mol), RPS7 (−302.44 kcal/mol), and GAPDH (−159.5 kcal/mol) (Fig. 5K). Considering the weak binding characteristic of GAPDH in the *in silico* analysis and lack of any interaction between GAPDH and HEV IRESl RNA in the pulldown assay, GAPDH was not considered a genuine interaction partner of HEV IRESl RNA. Therefore, binding energy of the HEV IRESl RNA and GAPDH complex was considered the minimum threshold value to select genuine HEV IRESl RNA-binding proteins (Fig. 5K, dotted line).

**RPL5 and DHX9 (RNA helicase A [RHA]) proteins are essential for the function of HEV IRESl.** To further examine the functional significance of host interaction partners of HEV IRESl, we used small interfering RNA (siRNA) against some of the host factors to deplete them in Huh7 cells, followed by measurement of HEV IRESl activity and g1-HEV replication. Treatment of Huh7 cells for 72 h with siRNAs against RPL5, RPL24, RPL26, RPL41, RPS3A, RPS7, and DHX9 depleted the corresponding proteins (Fig. 6A). Note that in the case of RPS3A siRNA, there was a significant reduction in GAPDH protein levels at 72 h (Fig. 6A). The effect of siRNA treatment on viability of Huh7 cells was measured at 72 h after treatment. RPS3A siRNA treatment led to a significant loss of

**FIG 3** Legend (Continued)

(LU⁻). Four random colonies from each cotransformant plate were replica plated onto medium lacking leucine, uracil, and histidine (LUH⁻) and supplemented with 5 mM or 10 mM 3-amino-1,2,4-triazole (3-AT). The same colonies were used in liquid *β*-galactosidase assays. Relative *β*-galactosidase units are plotted as mean ± SEM.

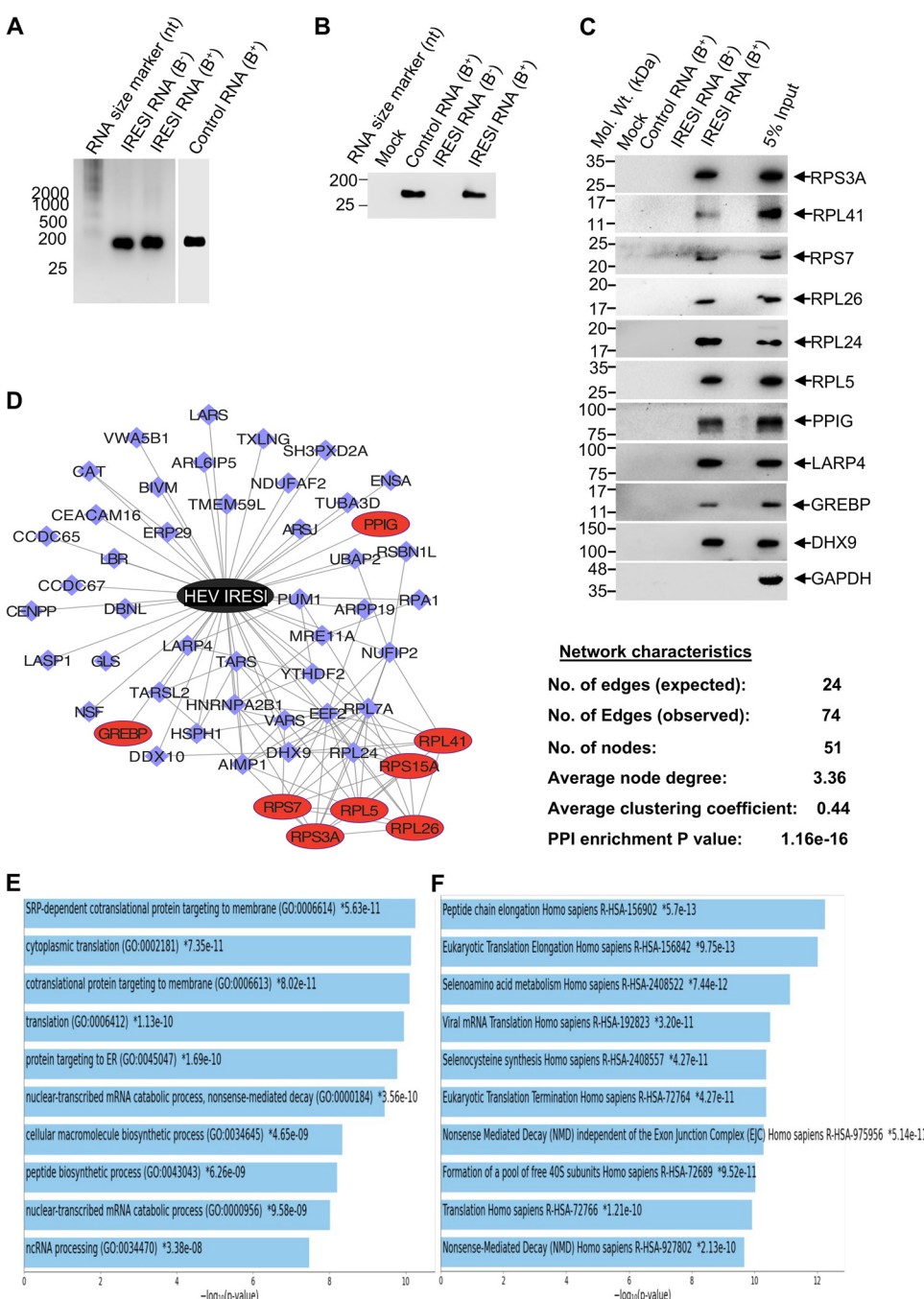

**FIG 4** Confirmation of HEV IRESl RNA-protein interactions by biotinylated RNA pulldown assay and construction of the RNA-protein interaction network. (A) Formaldehyde-agarose gel electrophoresis and ethidium bromide staining-mediated visualization of *in vitro*-transcribed and purified nonbiotinylated IRESl RNA (IRESl RNA [B⁻]), biotinylated IRESl RNA (IRESl RNA [B⁺]), and biotinylated control RNA (control RNA [B⁺]), as indicated. An RNA size marker was resolved in parallel (lane 1). (B) Formaldehyde-agarose gel electrophoresis and ethidium bromide staining-mediated visualization of eluates from streptavidin pulldown of samples containing control RNA (B⁺), IRESl RNA (B⁻), and IRESl RNA (B⁺), as indicated. (C) Western blotting of the indicated proteins using aliquots of eluates from biotinylated RNA pulldown of samples containing control RNA (B⁺), IRESl RNA (B⁻), and IRESl RNA (B⁺), as indicated; 5% input indicates that 5% of the cell lysate was used for incubation with the RNAs during the pulldown assay. (D) Schematic of the HEV IRESl RNA-protein interaction network. The black node denotes HEV IRESl RNA, and blue and red nodes denote host proteins identified by the RaPID and Y3H assays, respectively. Edges are represented by blue lines; PPI, protein-protein interaction. (E) Graphical representation of the top 10 biological processes (sorted by *P* values) enriched in the HEV IRESl RNA-protein interactome. (F) Graphical representation of the top 10 Reactome pathways (sorted by *P* values) enriched in the HEV IRESl RNA-protein interactome.

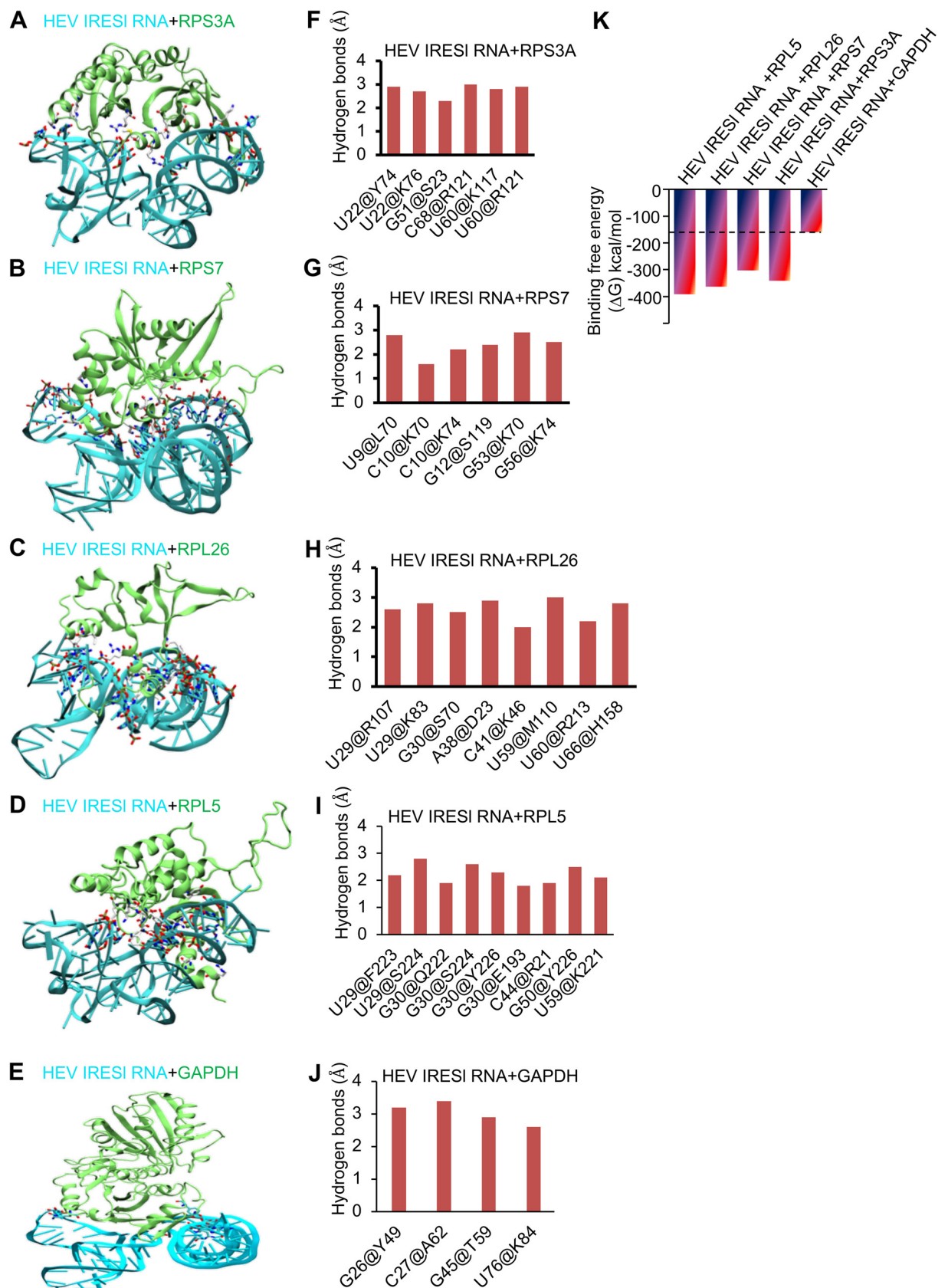

**FIG 5** *In silico* mapping of the interaction between HEV IRESI RNA and ribosomal proteins. (A) The interaction between HEV IRESI (cyan) and RPS3A (lime) is shown in the cartoon. The interacting nucleotides and amino acids are rendered in licorice and colored in atom-wise mode;

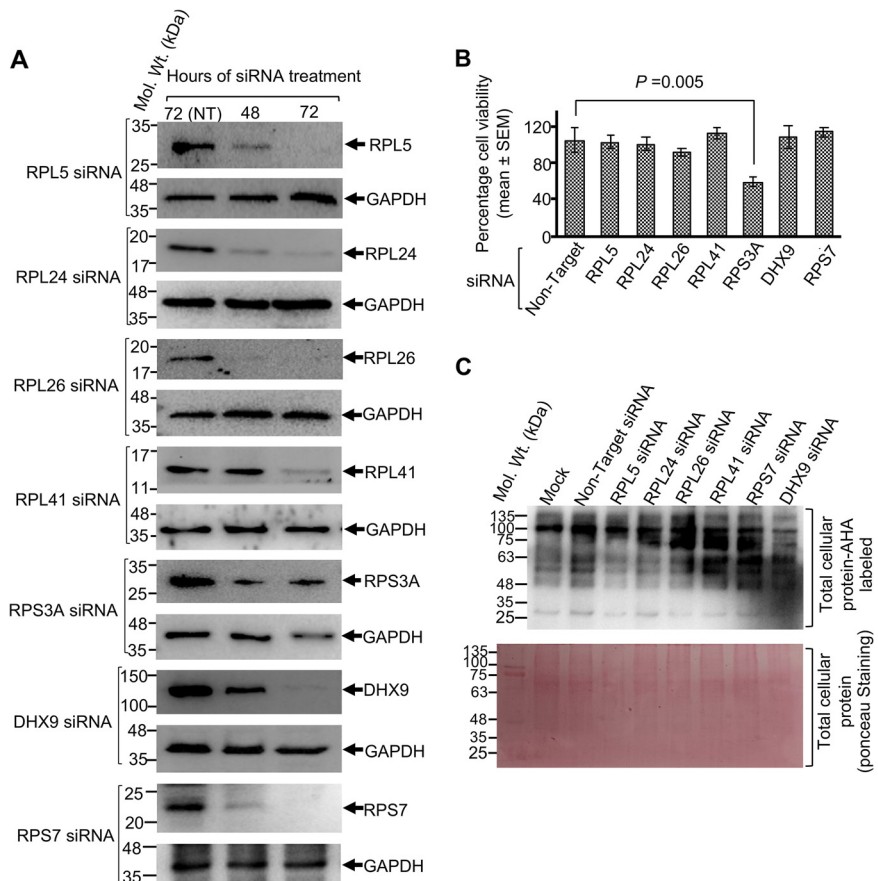

**FIG 6** Effects of siRNAs against selected ribosomal proteins and RNA helicase DHX9 on cell viability and global protein translation. (A) Western blotting of the indicated proteins after 48 and 72 h of treatment with the indicated siRNAs (lanes 2 and 3). Lane 1 shows protein from cells treated for 72 h with a nontarget (NT) siRNA. (B) Percent viability of Huh7 cells treated with the indicated siRNAs for 72 h. The value for the nontarget siRNA-treated cells was considered to be 100%, and other values were calculated relative to that. Values are shown as mean ± SEM of triplicate samples. (C) Top, Western blotting of AHA-labeled and biotinylated total cellular protein after 72 h of treatment with the indicated siRNAs (lanes 3 to 8). Lane 1 shows protein from mock-transfected cells, and lane 2 shows protein from cells transfected with nontargeting siRNA. Bottom, Ponceau staining of the blot shown on top.

cell viability (Fig. 6B). Therefore, RPS3A siRNA was excluded from further studies. Next, the effect of siRNA treatment on global translation was measured by transiently labeling the siRNA-transfected cells with L-azidohomoalanine (AHA). RPL5, RPL24, and DHX9 siRNAs partially reduced global translation, while RPL26, RPL41, and RPS7 siRNAs increased translation (Fig. 6C, top). A Ponceau-stained blot is shown for comparison of loading among different lanes (Fig. 6C, bottom). Note that none of the siRNAs completely abolished protein translation.

Next, a bicistronic reporter construct (pRL-FF/Luc dual-luciferase construct) was used to simultaneously measure HEV IRESl activity (by measuring firefly luciferase [Firefly-Luc] activity) and cap-dependent translation activity (by measuring *Renilla* luciferase [*Renilla*-Luc] activity) in siRNA-treated Huh7 cells, as described earlier (32) (Fig. 7A). Among the ribosomal

**FIG 5** Legend (Continued)
C, cyan/white; O, red; N, blue; S, yellow. (B) The interaction between HEV IRESl (cyan) and RPS7 (lime) is shown in the cartoon. The interacting nucleotides and amino acids are rendered as shown in A. (C) The interaction between HEV IRESl (cyan) and RPL26 (lime) is shown in the cartoon. The interacting nucleotides and amino acids are rendered as shown in A. (D) The interaction between HEV IRESl (cyan) and RPL5 (lime) is shown in the cartoon. The interacting nucleotides and amino acids are rendered as shown in A. (E) The interaction between HEV IRESl (cyan) and GAPDH (green) is shown in the cartoon. The interacting nucleotides and amino acids are rendered as shown in A. (F to J) The quantitative polar contacts between nucleotide and amino acids are measured for the indicated complexes and are shown in bar graphs (y axis, hydrogen bond distance; x axis, interacting pair of nucleotides@amino acids). (K) Binding free energy for the indicated complexes.

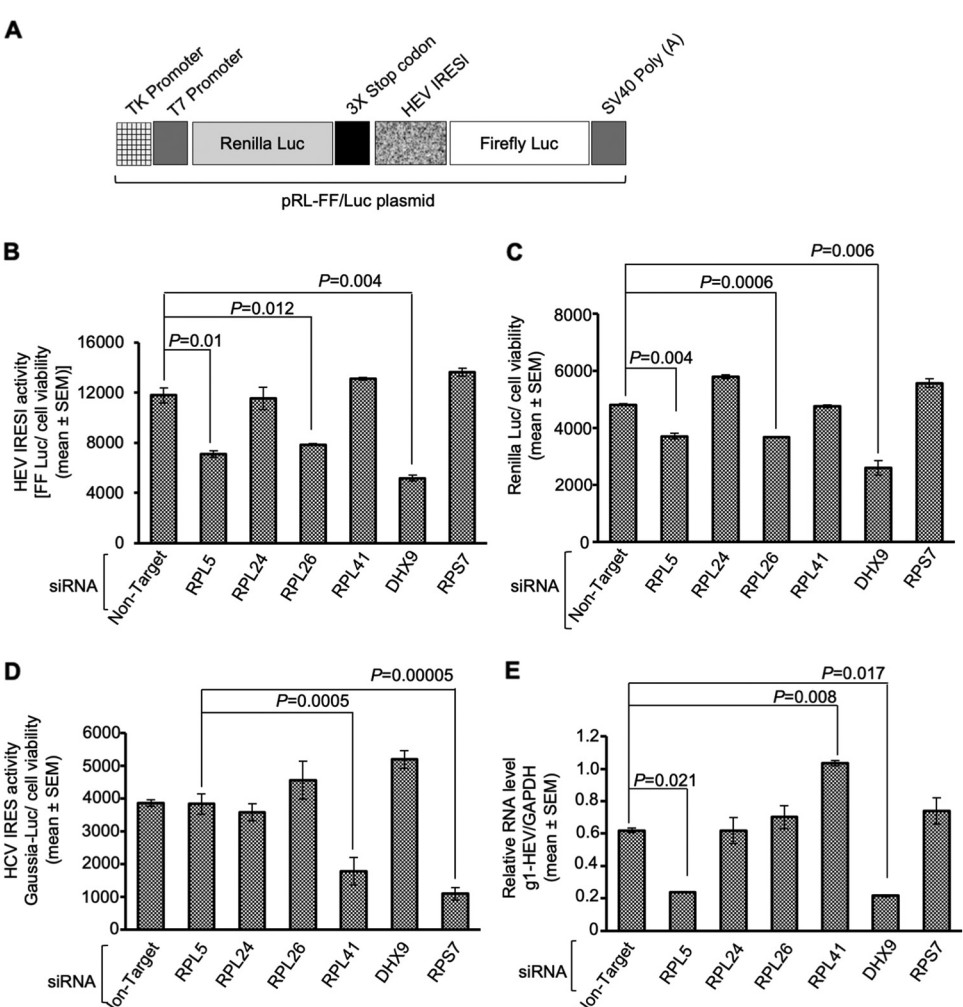

**FIG 7** Essential roles of the ribosomal protein RPL5 and RNA helicase DHX9 in mediating the function of the HEV IRESI element. (A) Schematic of the pRL-FF/Luc bicistronic reporter plasmid. (B) Firefly-Luc activity in Huh7 cells transfected with the indicated siRNAs and the pRL-FF/Luc plasmid for 72 h. The Firefly-Luc values were divided by cell viability and are represented as mean ± SEM of triplicate samples. (C) *Renilla*-Luc activity in Huh7 cell lysates used in B. The *Renilla*-Luc values were divided by cell viability and are represented as mean ± SEM of triplicate samples. (D) Gaussia-Luc activity in Huh7 cells transfected with the indicated siRNAs and the HCV-IRES-Gaussia-Luc cRNA for 72 h. The Gaussia-Luc values were divided by cell viability and are represented as mean ± SEM of triplicate samples. (E) RT-qPCR measurement of intracellular levels of g1-HEV genomic RNA in Huh7 cells transfected with the indicated siRNAs for 72 h. Values for g1-HEV RNA were normalized to *GAPDH* RNA and are represented as mean ± SEM of triplicate samples.

proteins, siRNAs against RPL5 and RPL26 significantly reduced the activity of both Firefly-Luc (HEV IRESI activity) and *Renilla*-Luc (cap-dependent translation activity) (Fig. 7B and C). Lack of RPL24 and RPL41 did not affect Firefly-Luc or *Renilla*-Luc activity, but DHX9 siRNA significantly decreased the activity of both Firefly-Luc and *Renilla*-Luc (Fig. 7B and C). We also measured the effects of the different siRNAs on the activity of HCV IRES by using a Gaussia luciferase (G-Luc) reporter, which is under the control of the HCV IRES element. Among the ribosomal proteins, siRNAs against RPL41 and RPS7 significantly reduced the activity of HCV IRES (Fig. 7D). DHX9 siRNA had no effect on HCV IRES activity (Fig. 7D). These data show the distinct functions of different ribosomal proteins and DHX9 during different modes of protein translation.

Next, g1-HEV-expressing Huh7 cells were treated with different siRNAs, followed by measurement of the level of viral sense-strand RNA. Lack of RPL5 and DHX9 proteins significantly reduced the level of g1-HEV RNA (Fig. 7E). However, lack of RPL24 and RPL26 did not affect g1-HEV RNA levels, and lack of RPL41 marginally enhanced the level of g1-HEV RNA (Fig. 7E).

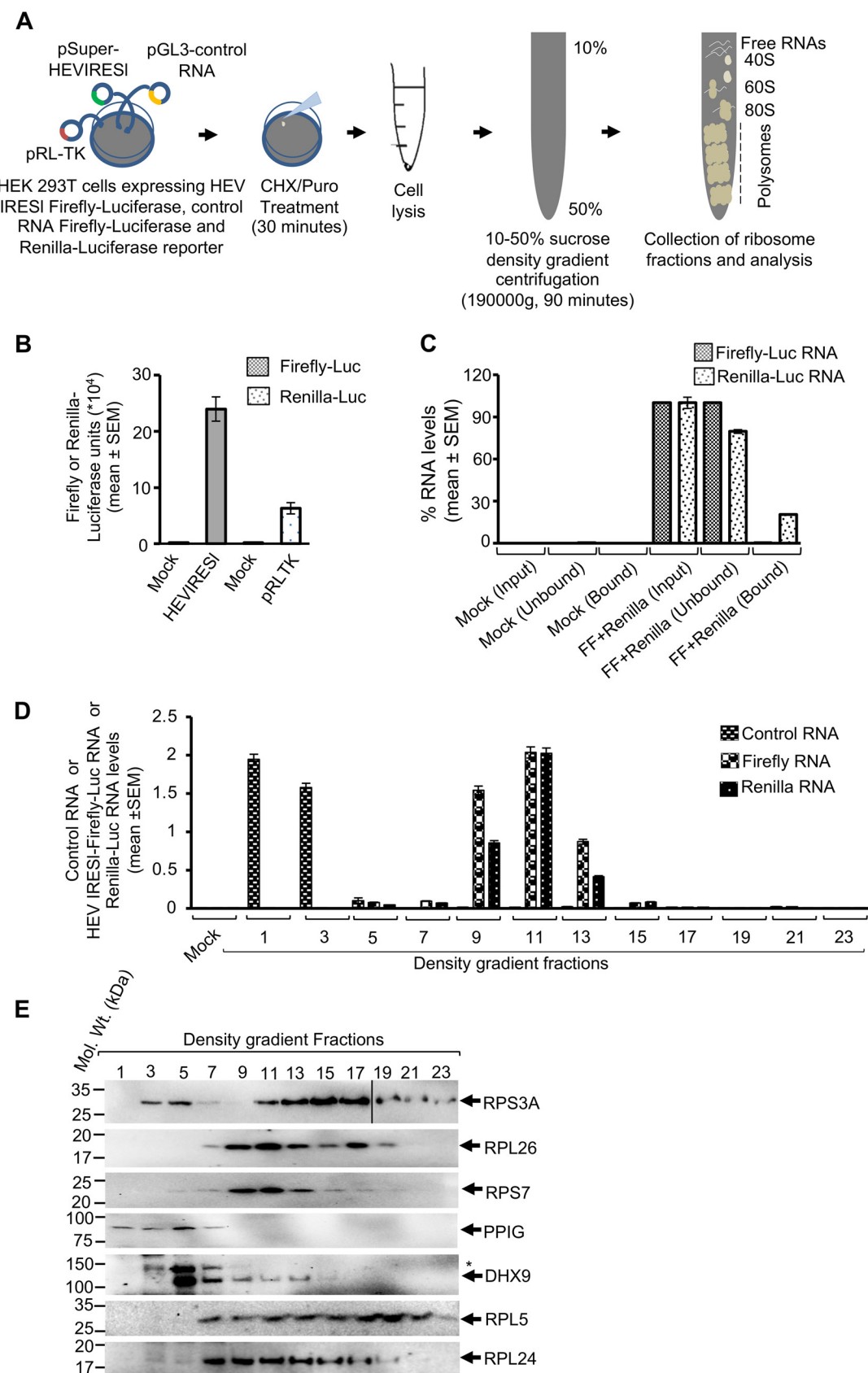

**FIG 8** The HEV IRESl element associates with actively translating ribosomes. (A) Schematic of the ribosome fractionation assay; CHX, cycloheximide; Puro, puromycin. (B) Measurement of Firefly-Luc and *Renilla*-Luc activities in Huh7 cells cotransfected with

**HEV IRESl RNA is associated with polysomes in translationally active cells.** A ribosomal fractionation assay was performed to verify the localization of the HEV IRESl element in actively translating ribosomes (Fig. 8A). A construct expressing the Firefly-Luc reporter under the control of HEV IRESl in a cap-independent manner (pSuper-HEV IRESl) was used to distinctly identify the presence of HEV IRESl in actively translating ribosomes. Note that HEV IRESl RNA transcription is driven by the H1 RNA polymerase III promoter in the pSuper vector. A *Renilla*-Luc reporter driven by cap-dependent translation was used as a positive control for the assay (pRL-TK). A control RNA (pGL3 control RNA) was used to check specificity of the polysome fractions. HEK293T cells were cotransfected with the pSuper-HEV IRESl and pRL-TK plasmids. Forty-eight hours posttransfection, cells were lysed, and Firefly-Luc and *Renilla*-Luc activity was measured. As expected, significant activity of both Firefly-Luc and *Renilla*-Luc was observed (Fig. 8B). To verify if Firefly-Luc was produced through a cap-independent process, total RNA was isolated from the cotransfected cells, followed by immunoprecipitation of capped RNAs using the anti-$m^7G$ cap antibody, which specifically recognizes the $m^7G$ in the cap structure (43). Levels of Firefly-Luc- or *Renilla*-Luc-encoding RNA were quantified by real-time quantitative PCR (RT-qPCR). As expected, *Renilla*-Luc RNA was detected in the anti-$m^7G$ cap antibody-immunoprecipitated sample, whereas Firefly-Luc RNA was not detectable in aliquots of the same sample (Fig. 8C). By contrast, Firefly-Luc RNA was detected in the unbound fraction only (Fig. 8C).

Next, cytoplasmic fractions of HEK293T cells cotransfected with the pSuper-HEV IRESl and the pRL-TK plasmids were subjected to linear (10 to 50%) sucrose density gradient centrifugation, followed by measurement of both Firefly-Luc and *Renilla*-Luc RNAs in different fractions. Cells were lysed after treatment with cycloheximide (CHX) or puromycin (Puro), which stalls translating ribosomes or prevents polysome assembly, respectively (44, 45). Ribosome profiles of cycloheximide- and puromycin-treated cells showed an expected pattern, in agreement with earlier studies (Fig. S4A) (45). Total RNA was isolated from aliquots of the alternate fractions from the density gradient samples and was resolved by formaldehyde-agarose gel electrophoresis, followed by visualization of 28S and 18S RNA by ethidium bromide staining. As expected, 28S and 18S RNA were primarily present in the fractions corresponding to polysomes in the cycloheximide-treated samples (Fig. S4B, top) and were significantly reduced in the puromycin-treated samples (Fig. S4B, bottom). RT-qPCR revealed that in cycloheximide-treated cells, both Firefly-Luc and *Renilla*-Luc RNA were predominantly detected in fractions 9 onward, indicating their association with polysomes (Fig. 8D). On the other hand, control RNA was mostly present in fractions 1 and 3 (representing free RNA), indicating its inability to associate with the polysome, which further confirms the specificity of the ribosome fractionation assay (Fig. 8D). Next, aliquots of the density gradient fractions were used for Western blotting of the different ribosomal proteins DHX9 and PPIG. In cycloheximide-treated cells, RPS3A, RPL26, RPS7, RPL5, and RPL24 were predominantly present in the polysome fractions (Fig. 8E). DHX9 was found in both 40S-80S ribosome-containing fractions and polysome fractions (Fig. 8E). PPIG was predominately found in the free mRNA- or 40S-80S ribosome complex-containing fractions (Fig. 8E). PPIG was used as a control to monitor purity of the polysome fractions as it is not known to be present in polysomes. Note that PPIG is a peptidyl-prolyl isom-

**FIG 8** Legend (Continued)

the pSuper-HEV IRESl and the pRL-TK plasmids. Mock shows pSuper vector-transfected Huh7 cells processed in parallel. Values are represented as mean $\pm$ SEM of triplicate samples. (C) RT-qPCR analysis of Firefly-Luc and *Renilla*-Luc RNAs obtained by the RNA immunoprecipitation assay using anti-$m^7G$-cap antibody; Mock, RNA from the pSuper vector-transfected Huh7 cells; FF+Renilla, RNA from the Huh7 cells cotransfected with the pSuper-HEV IRESl and pRL-TK plasmids; Input, amount of RNA used for RNA immunoprecipitation; Unbound, RNA isolated from the supernatant of immunoprecipitation samples before washing; Bound, RNA isolated from the elution fraction of the immunoprecipitation samples. Values are shown as mean $\pm$ SEM of triplicate samples. (D) RT-qPCR analysis of control RNA, Firefly-Luc RNA, and *Renilla*-Luc RNA levels in aliquots of total RNA isolated from sucrose density gradient fractions of Huh7 cells cotransfected with the pSuper-control RNA plasmid, pSuper-HEV IRESl plasmid, and pRL-TK plasmid and treated with cycloheximide. Values are shown as mean $\pm$ SEM of triplicate samples. (E) Western blotting of the levels of indicated proteins present in the density gradient fractions as shown in D. The asterisk (*) indicates a nonspecific band detected by the anti-DHX9 antibody.

erase of the cyclophilin family, which catalyzes the conversion between *cis* and *trans* isomers of proline (46). Collectively, these findings confirm that HEV IRESl RNA is complexed with actively translating ribosomes.

The importance of HEV IRESl RNA-binding translation-related proteins in driving the association between HEV IRESl RNA and polysomes was further assessed by a ribosome fractionation assay in Firefly-Luc- and *Renilla*-Luc-expressing HEK293T cells treated with different siRNAs and cycloheximide. Fractions 1, 5, 9, and 13 from the density gradient centrifugation samples were analyzed to quantify the levels of Firefly-Luc and *Renilla*-Luc RNAs as free RNA (fraction 1), in the 40S-60S ribosome complex (fraction 5), or in the polysome complex (fractions 9 and 13). Treatment of cells with siRNAs against RPL5, RPL26, and DHX9 significantly reduced HEV IRESl RNA association with polysomes, whereas siRNAs against RPL24, RPL41, and RPS7 had no effect (Fig. 9A). However, in the case of *Renilla*-Luc RNA, in addition to RPL5 and RPL26, treatment with RPL24 siRNA also significantly reduced the association between *Renilla*-Luc RNA and polysomes. Further, treatment with DHX9, RPL41, or RPS7 siRNA did not affect the association between *Renilla*-Luc RNA and polysomes (Fig. 9B). Next, aliquots of the above samples were analyzed for detection of the RPL5 protein (an indicator of polysome-containing fractions). As expected, RPL5 was predominately detected in the polysome-containing fractions (Fig. 9C, nontarget siRNA). Treatment with RPL5 siRNA significantly reduced its level, eventually leading to a reduction in the level of polysomes. Further, knockdown of RPL24 and RPS7 also decreased the level of polysomes, whereas RPL26, RPL41, and DHX9 siRNA treatment had no effect on polysome levels (Fig. 9C). Collectively, these findings confirm the association of HEV IRESl RNA with actively translating ribosomes and suggest that RPL5 and DHX9 help in HEV IRESl-mediated translation by holding the RNA in polysomes, whereas DHX9 is dispensable for holding the *Renilla*-Luc RNA (cap-dependent translation) in polysomes.

## DISCUSSION

While investigating the cause of poor replication of g1-HEV in Huh7 cells, we observed better viral replication after treatment of cells with ER stress-inducing compounds such as tunicamycin and thapsigargin. Subsequently, we identified a fourth ORF in the viral genome, which produces the ORF4 protein that plays a key role in assembly of the viral replication complex (32). We also showed that ORF4 is synthesized by an IRES-like (HEV IRESl) element. Sequence analysis of HEV IRESl indicated its weak homology with the equine rhinitis A virus (ERAV) IRES (a type II IRES) (32, 47). The HEV IRESl element is remarkably smaller in size than other canonical IRES elements, and it requires only 87 nucleotides to drive cap-independent translation of the ORF4 protein in its native context as well as drive the translation of a heterologous protein, such as Firefly-Luc, when inserted into a bicistronic reporter construct. By contrast, the function of known IRES elements depends on double the number of nucleotides or more. The smaller size and relatively simple secondary structure of the HEV IRESl element offer the advantage of dissecting the molecular mechanisms that are crucial for initiation of cap-independent internal translation. Therefore, the current study was undertaken to further substantiate the claim that the HEV IRESl element is a bona fide internal ribosome entry site.

Two independent approaches were used to identify the host interaction partners of HEV IRESl RNA: the RaPID assay and the Y3H assay. The RaPID assay identifies stable and transient interactions of all proteins associated with the RNA of interest that include both direct and indirect interaction partners of the RNA. Several controls were used to avoid/subtract nonspecific interaction partners, thus ensuring the generation of high-confidence mass spectrometry data for downstream studies. For example, the good quality of our HEV IRESl RaPID assay LC-MS/MS data is evident from the fact that comparison of the HEV IRESl-interacting proteins with the severe acute respiratory syndrome coronavirus 2 (SARS-CoV-2) 5′- and 3′-UTR RNA-interacting proteins (reported by our laboratory using the same technique) identified only RPL7a as a common protein (34). RPL7a is an essential component of the large subunit of the ribosome and is known to directly bind RNA through its two distinct RNA-binding domains (48). The presence of RPL7a in both data

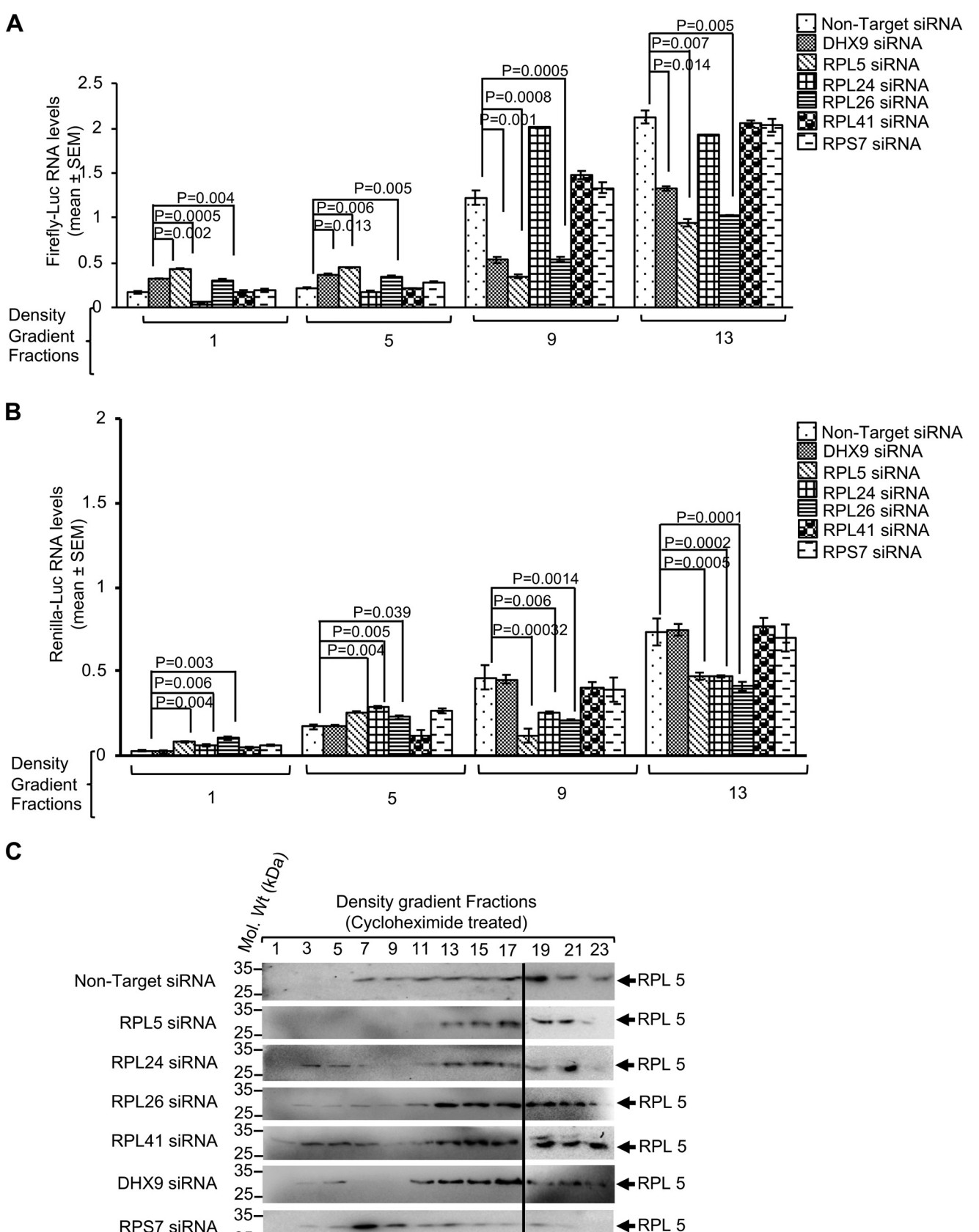

**FIG 9** RPL5, RPL26, and RNA helicase A are important for association of the HEV IRESI element with the polysome. (A) RT-qPCR analysis of the Firefly-Luc RNA level in density gradient fractions 1, 5, 9, and 13 of Huh7 cells transfected with the indicated siRNAs and the pSuper-HEV IRESI and pRL-TK

sets seems to be in agreement with its required function rather than being an experimental artifact.

A limitation of the RaPID assay is attributed to its dependence on biotinylation of lysines exposed on the surface of RNA-binding proteins present within a distance of 10 nm from the BirA ligase. Thus, it is possible to lose some of the RNA-binding proteins if they are buried deep inside the complex and are inaccessible to BirA ligase for biotinylation. Therefore, a conventional Y3H assay-based human liver cDNA library screening was conducted in parallel to identify the interaction partners of HEV IRESl RNA. The Y3H assay offers the advantage of identifying the direct interaction partners of a bait RNA and identified ribosomal proteins (RPL5, RPL26, RPL41, RPS7, RPS15A, and RPS3A), a known DNA-binding protein (GREBP), and a cyclophilin family protein (PPIG) as interaction partners of HEV IRESl (46, 49). Specificity of the Y3H data was further ensured by assessing the interaction of the eight HEV IRESl RNA-interacting proteins with two well-known IRES sequences, the FMDV IRES (type II IRES) and the HCV IRES (type IV IRES). Distinct patterns of interaction were observed for these RNA sequences, supporting the specificity of the interactions detected in the Y3H assay. Although both RaPID and Y3H assays identified specific interaction partners of HEV IRESl, no common proteins were detected in the two assays. A search of the initial RaPID data of the HEV IRESl sample identified RPL26 as a common protein present in both RaPID and Y3H data sets. However, RPL26 was not present in the final list of HEV IRESl interaction partners, as it had a PEP score and Prot score of 7 and 29, respectively, whereas the cutoff threshold was set to 17 and 30, respectively. Note that RPL26 was not present in the negative-control sample RaPID data.

To further confirm the interaction between HEV IRESl RNA and host proteins, an *in vitro* biotinylated RNA pulldown assay and molecular docking analysis were performed for some of the HEV IRESl-interacting proteins, which supported their interaction. Docking analysis also demonstrated that the complex between HEV IRESl and RPL5 was more stable than the others. Moreover, comparative interaction mapping showed that the interaction between the HEV IRESl element and the ribosomal proteins primarily involved SL A (nucleotides U9, C10, G12, and U22), SL B (nucleotides U29, G30, A38, C41, C44, and G50), and the unpaired region between SL B and SL C (G51, G53, G56, U59, U60, U66, and C68) of the former (refer to the HEV IRESl secondary structure in Fig. 1A). In summary, multiple validation steps ensured that the HEV IRESl RNA interaction partners identified in our experiments are of high confidence.

Bioinformatic analysis of the 51 HEV IRESl RNA-binding proteins revealed enrichment of proteins involved in translation. Eight ribosomal proteins and six translation regulatory factors interact with HEV IRESl RNA, in support of its function as a translation initiation site. Four proteins identified in our study are known to interact with the IRESs of other viruses (DHX9, RPA1, RPL26, and RPS3A).

Silencing of RPL5 significantly inhibited HEV IRESl activity as well as g1-HEV replication. RPL5 strongly interacts with HEV IRESl, weakly interacts with HCV IRES, and does not interact with FMDV IRES in the Y3H assay. RPL5 showed the lowest binding free energy ($\Delta G$) for interaction with HEV IRESl RNA *in silico*. RPL5 is also a component of the polysome, and lack of RPL5 significantly affected polysome stability and dissociated both HEV IRESl and *Renilla*-Luc RNAs from the polysome. Hence, it is clear that RPL5 is an essential factor for both cap-dependent and cap-independent translation. RPL26 showed a pattern similar to RPL5 by inhibiting HEV IRESl-Luc and *Renilla*-Luc activities. Moreover, the HEV IRESl Firefly-Luc and *Renilla*-Luc RNA association with polysomes was significantly reduced after treatment of cells with RPL26 siRNA. However,

**FIG 9** Legend (Continued)
plasmids for 72 h and treated with cycloheximide for 30 min. Values are shown as mean ± SEM of triplicate samples. (B) RT-qPCR analysis of *Renilla*-Luc RNA levels in density gradient fractions 1, 5, 9, and 13 of Huh7 cells transfected with the indicated siRNAs and the pSuper-HEV IRESl and pRL-TK plasmids for 72 h and treated with cycloheximide for 30 min. Values are shown as mean ± SEM of triplicate samples. (C) Western blotting of RPL5 protein levels in the indicated density gradient fractions of Huh7 cells transfected with the indicated siRNAs and the pSuper-HEV IRESl and pRL-TK plasmids for 72 h and treated with cycloheximide for 30 min.

RPL26 differed from RPL5 in the following parameters: it bound moderately with HEV IRESl RNA both in Y3H assays and in *in silico* analyses, a lack of RPL26 did not affect g1-HEV replication, and a lack of RPL26 did not affect polysome stability. Earlier studies have shown that the lack of RPL26 does not affect cap-dependent translation in *Saccharomyces cerevisiae*, and RPL26 is a known IRES transacting factor (ITAF), which has been shown to control DNA damage-induced cap-independent translation of *p53* mRNA (50–52). Based on our data and available literature, we propose that RPL26 may be involved in mediating the functional efficiency of HEV IRESl, although it is not an essential factor. Ablation of both RPL5 and RPL26 may show a more profound effect on HEV IRESl activity. Further studies are required to clarify the role of RPL26 in mediating the function of HEV IRESl. Silencing of RPL24 and RPL41 did not inhibit either HEV IRESl or *Renilla*-Luc activity. Lack of RPL24 reduced the stability of the polysomes, which is in agreement with earlier reports (53). However, lack of RPL24 did not affect the association between HEV IRESl RNA and polysomes, although it reduced the association of *Renilla*-Luc RNA with polysomes, indicating its importance in cap-dependent translation. Lack of RPL41 did not affect polysome stability or HEV IRESl and *Renilla*-Luc RNA association with polysomes. Interestingly, lack of RPL41 significantly increased g1-HEV replication, possibly due to extratranslational functions of RPL41. Note that RPL41 knockdown has been shown to arrest the cell cycle in the $G_2$/M phase, and it acts by stabilizing the microtubules (54). Our earlier study showed that HEV replication is significantly higher in the $G_2$/M phase of the cell cycle (55).

Apart from the ribosomal proteins, DHX9/RNA helicase A (RHA), HNRNPA2/B1, YTHDF2, EEF2, and LARP4 were also identified as interaction partners of HEV IRESl RNA. RHA is known to associate with the IRESs of FMDV and HCV and acts as an ITAF, promotes translation from the IRES of retroviral RNA and *JUND* mRNA, and regulates the replication of several RNA viruses, including the porcine reproductive and respiratory syndrome virus, bovine viral diarrhea virus, classical swine fever virus, and dengue virus (56–59). RHA binds to the 3′ UTR of HEV; however, significance of the interaction is not known (60). RHA is a member of the DEx/H-box family of superfamily II (SF2) helicases, which unwind duplex RNA that possesses a 3′-single-stranded RNA tail by hydrolyzing the nucleoside triphosphates (61, 62). In our experiments, lack of RHA significantly reduced the activities of both HEV IRESl and *Renilla*-Luc as well as g1-HEV replication. RHA was found to be associated with 40S-80S ribosomes and polysomes; however, its absence did not affect polysome stability. Lack of RHA resulted in reduced recruitment of HEV IRESl RNA to the polysomes; however, association of *Renilla*-Luc RNA with polysomes was not affected. Although polysome-associated mRNAs are generally considered to be translationally active, earlier studies have shown that some polysome-associated mRNAs, such as *ACTB* mRNA, are not translated during mitosis (63). The mechanism of translation inhibition of polysome-associated *ACTB* mRNA remains to be identified; however, the authors proposed that such control could help in rapid resumption of *β*-actin synthesis after exit from mitosis, and attenuation of translation termination could be a possible mechanism (63). It should also be noted that pharmacological inhibitors of protein synthesis, such as cycloheximide, stabilize polysome-associated mRNAs. Inhibition of polysome-associated *Renilla*-Luc mRNA translation in the absence of RHA might be attributed to an analogous mechanism. Further experiments are required to identify the actual mechanism. Nevertheless, our current data clearly support a role of RHA in controlling the activity of HEV IRESl. HNRNPA2B1 is reported to be involved in mRNA processing and has been shown to bind the 3′ UTR of *CDK6* and recruit RHA, leading to microRNA (miRNA)-mediated silencing of CDK6 (64). An earlier study showed that HNRNPA2B1 binds to the genomic and subgenomic promoters of HEV and plays an essential role in g1-HEV replication. Further, intracellular localization of HNRNPA2B1 is altered in HEV-infected cells (65). YTHDF2 is known to accelerate mRNA decay by targeting mRNAs to cytoplasmic processing bodies (P-bodies) and by recruiting the CCR4-NOT deadenylase complex (66, 67). Eukaryotic elongation factor 2 (EEF2) is a GTP-binding translation elongation factor that plays an essential role in translation by promoting GTP-dependent translocation of the ribosome (68). LARP4 binds to the poly(A) tract of mRNA, associates with the 40S ribosomal subunit and polysomes, and

plays a role in translation regulation by preventing deadenylation at the poly(A) tail (69–71). The presence of these factors in the HEV IRESl RNA-associated protein complex suggests their potential role(s) in regulating HEV IRESl-mediated translation. Further studies are warranted to clarify their mode of action.

An important aspect of HEV IRESl function pertains to upregulation of its activity by ER stress-inducing agents in g1-HEV-infected cells (32). The RaPID assay identified four proteins involved in the ER stress response pathway, including heat shock protein 105 kDa (HSP 105), the luminal ER protein of 29 kDa (ERp29), γ-taxilin, and Pumilio homolog 1 (PUM1), as interaction partners of HEV IRESl. HSP 105 is known to regulate ER stress-induced caspase-3 activation (72). ERp29 is induced by ER stress, is ubiquitously expressed, and is implicated in the biosynthesis and trafficking of many proteins (73). γ-Taxilin belongs to the syntaxin-binding protein family, interacts with ATF4, and regulates the ER stress response (74). PUM1 is known to be phosphorylated by the IRE1 kinase under conditions of ER stress and binds spliced *XBP1* RNA and prevents its degradation (75). Following ER stress, interaction between PUM1 and HEV IRESl might be protecting the latter from degradation by the regulated IRE1-dependent decay (RIDD) pathway.

In conclusion, we have identified 51 host proteins that associate with HEV IRESl RNA in mammalian cells. Four of the identified proteins have been reported to associate with known IRES elements of viral and cellular origin, and two of the identified proteins are known ITAFs. Several ribosomal proteins, translation regulatory factors, and components of the tRNA synthetase complex were found to be associated with HEV IRESl RNA. Taken together, the above data further support the notion that HEV IRESl is a bona fide translation initiation site capable of driving cap-independent translation. These findings significantly advance our understanding of the molecular mechanisms that control synthesis of the ORF4 protein in g1-HEV-infected cells. Moreover, considering the small size and simplicity of the secondary structure of the HEV IRESl element compared to other well-known IRES elements, it is a useful tool for recombinant protein expression as well as for investigation of the molecular details of the cap-independent internal translation process.

## MATERIALS AND METHODS

**Plasmids and reagents.** The RNA motif plasmid cloning backbone (pRMB; Addgene plasmid 107253, http://n2t.net/addgene:107253, RRID: Addgene_107253) and the BASU RaPID plasmid (Addgene plasmid 107250, http://n2t.net/addgene:107250, RRID:Addgene_107250) were gifted by Paul A. Khavari (School of Medicine, Stanford University, CA, USA) and were reported earlier (29). To clone the HEV IRESl in the pRMB vector, nucleotides 2701 to 2787 of the g1-HEV genome (GenBank ID: AF444002.1) were PCR amplified from the pSK HEV2 plasmid (32) with the primers IRESl87 BASU CBB FP (5′-GCTTGGTGCCGATCGGTCCC-3′) and IRESl87 BASU CBB RP (5′-AGCTCATCTGGCAGCAAGCTCAG-3′) and were ligated with the pRMB vector backbone generated by BSMB1 restriction enzyme digestion, followed by blunting. For hybrid RNA generation for the Y3H assay, HEV IRESl, FMDV IRES, and HCV IRES sequences were PCR amplified from the pSKHEV2 plasmid, pVitro2-neo-mcs plasmid (InvivoGen, San Diego, CA, USA), and S52/SG-Feo(AI) replicon (gifted by Charles M. Rice [76]), respectively, using the following primers: HEV IRESl FP (5′-ACTATCTCGAGGGTGCCGA TCGGTCCC-3′), HEV IRESl RP (5′-ACTATCCATGGCATCTGGCAGCAAGCTCAG-3′), FMDV IRES FP (5′-ACTATCT CGAGAGCAGGTTTCCCCAATGACACA-3′), FMDV IRES RP (5′-ACTATCCATGGAAAGGAAAGGTGCCGACCTCCG-3′), HCV IRES FP (5′-ACTATCTCGAGACCTGCCTCTTACGAGGC-3′), and HCV IRES RP (5′-ACTATCCATGGTTCTT TGAGGTTTAGGAAGTGTGCT-3′). PCR products were digested with the XhoI restriction enzyme and ligated with the XhoI- and SmaI-digested p3HR2 vector backbone. To clone the negative-sense HEV IRESl (HEV IRESl⁻) fragment in the p3HR2 vector, 87 nucleotides of the IRESl sequence were PCR amplified from the pSK HEV plasmid using HEV IRESl FP and HEV IRESl RP primers and ligated with the SmaI-digested p3HR2 vector. To clone the HEV IRESl sequence in the pSuper vector, pRLFF/Luc87 and pSuper plasmids were restriction digested with BamHI and KpnI, respectively, followed by treatment with Klenow (end-filled by Klenow) and digestion of both plasmids with XhoI and ligation. To clone the control RNA (nucleotides 618 to 738 in the g1-HEV genome) into the pGL3 vector (for use in the ribosome fractionation assay), the pGL3 vector was digested with HindIII and NcoI followed by blunting to generate the vector backbone. Next, control RNA sequence was amplified by PCR from pSKHEV2 (GenBank ID: AF444002.1) using the primers Control RNA FP (5′-GCCCCCTGGCACATA-3′) and Control RNA RP (5′-GAGCGCAGGTTGGAAA-3′). Linearized pGL3 vector backbone and the control RNA sequence insert were ligated, and positive clones were selected by restriction digestion. For biotinylated RNA synthesis, HEV IRESl and control RNA sequences were PCR amplified using primers as described above and cloned into the pJET1.2 vector following the manufacturer's instructions (Thermo Scientific, MA, USA). For generating the clone containing the HCV IRES element upstream of the Gaussia luciferase (G-Luc) coding region, a circular RNA (cRNA) vector backbone was used (77). A DNA cassette containing the sequence of the T7 promoter, the group I catalytic intron element from the Anabaena

pre-tRNA, the HCV-IRES sequence (described above for cloning into the p3HR2 vector), the G-Luc coding sequence, and the poly(A) sequence was commercially synthesized (Genscript, USA) and cloned into the pUC57 vector. The resulting plasmid was named pUC57 cRNA HCV IRES-G-Luc. All clones were confirmed by DNA sequencing. All plasmids are available upon request.

A human liver cDNA library was obtained from Clontech (CA, USA). The dual-luciferase reporter assay kit and CellTiter96 Aqueous one solution cell proliferation assay kits were from Promega (Madison, WI, USA). Nontargeting siRNA (D-001810-10-20), human RPL5 (6125) siRNA (L-013611-00-0005), human RPL26 (6154) siRNA (L-011132-01-0005), human RPL24 (6152) siRNA (L-011144-02-0005), human RPL41 (6171) siRNA (L-011160-01-0005), human RPS3A (6188) siRNA (L-013607-00-0005), and human DHX9 (1660) siRNA (L-009950-00-0005) were from GE Healthcare Dharmacon (CO, USA). Antibodies against RPL5 (A303-933A) and RPL26 (A300-686A) were from Bethyl Laboratories (TX, USA). Anti-RPL24 (ITT09128), anti-RPL41 (ITT09136), and anti-DHX9 (ITT12266) antibodies were from Geno Technology, Inc. (St. Louis, MO, USA). Anti-RPS3A (STJ28158), anti-RPS7 (STJ28814), and anti-PPIG (STJ191133) antibodies were from St. John's Laboratory (London, UK). Anti-GAPDH (sc-25778) antibody was from Santa Cruz Biotechnology (TX, USA). Goat anti-rabbit IgG-horseradish peroxidase (HRP; 4030-05) was from Southern Biotech (AL, USA). 3-Amino-1,2,4-triazole (3-AT), $ortho$-nitrophenyl-$\beta$-galactoside (ONPG), cycloheximide, and puromycin were from Sigma (MO, USA). Cycloheximide and puromycin were used at working concentrations of 100 $\mu$g/mL and 0.5 $\mu$g/mL, respectively.

**Mammalian cell culture, transfection, *in vitro* transcription, global translation measurement, and cell viability assays.** HEK293T cells were obtained from ATCC (VA, USA). Huh7 cells were cultured as described previously (32). Cells were maintained in Dulbecco's modified Eagle medium (DMEM) containing 10% fetal bovine serum (FBS) and 50 IU/mL penicillin and streptomycin at 37°C in a 5% $CO_2$ incubator. Cells were maintained in antibiotic-free medium before starting the experiments. For plasmid transfection, cells were seeded at 70 to 80% confluence in DMEM + 10% FBS and incubated overnight at 37°C and 5% $CO_2$. The next day, cells were transfected with the desired plasmids using Lipofectamine 2000 transfection reagent (Thermo Scientific, MA, USA) at a 1:1 ratio, following the manufacturer's instructions. Six to 8 h posttransfection, culture medium was replaced with fresh DMEM + 10% FBS.

For experiments involving siRNA-mediated gene silencing, cells were seeded at 70 to 80% confluence on 12-well tissue culture dishes and incubated overnight at 37°C with 5% $CO_2$. The next day, 25 nmol siRNA was transfected into each well using 0.35 $\mu$L of Dharmafect transfection reagent, following the manufacturer's instructions (GE Healthcare Dharmacon, Inc., CO, USA). Twelve hours posttransfection, the culture medium was replaced with fresh medium (DMEM + 10% FBS), and cells were maintained at 37°C in a 5% $CO_2$ incubator.

HEV genomic RNA was *in vitro* synthesized as capped RNA using an mMessage mMachine kit (Thermo Scientific, MA, USA), as previously described (32). The cRNA HCV IRES-G-Luc precursor was synthesized by *in vitro* transcription from the linearized plasmid DNA template using a MEGAscript T7 transcription kit (Thermo Scientific, MA, USA). After *in vitro* transcription, the reaction was treated with DNase I (Thermo Scientific, MA, USA), and transcripts were circularized by supplementing the sample with GTP, as previously described (77). Size and integrity of the *in vitro*-synthesized RNA was monitored by formaldehyde-agarose gel electrophoresis. *In vitro*-synthesized RNAs were transfected using Lipofectamine 2000 transfection reagent at a 1:1 ratio, following the manufacturer's instructions (Thermo Scientific, MA, USA). Six to 8 h posttransfection, the culture medium was replaced with fresh medium (DMEM + 10% FBS) and was maintained until further manipulation.

For biotinylated RNA pulldown assays, pJET1.2 HEV-IRESl and pJET1.2 control RNA plasmids were linearized by restriction digestion with NcoI and column purified. One microgram of purified linearized DNA was used for each *in vitro* transcription reaction using a MAXIscript T7 transcription kit, following the manufacturer's instructions (Thermo Scientific, MA, USA). Biotin-16-UTP (0.2 mM; Sigma-Aldrich, 11388908910) and unlabeled-UTP (0.3 mM) were added to the reaction mixture to produce biotinylated RNA. Nonbiotinylated RNAs were transcribed in parallel in the presence of 0.5 mM unlabeled UTP. *In vitro*-transcribed RNA was purified as per the manufacturer's instructions for the MAXIscript kit.

Global translation was measured by transient labeling of cells with L-azidohomoalanine, as described earlier (55). Cell viability was measured using a commercially available kit (CellTiter 96 aqueous one solution cell proliferation assay, Promega, Madison, WI, USA), which uses a tetrazolium salt-based colorimetric assay. Details are as described in Nair et al. (32).

**Yeast three-hybrid (Y3H) assay and screening of the human liver cDNA library.** A GAL4-based Y3H assay system, gifted by Marvin Wickens (University of Wisconsin, Madison, WI, USA), was used to determine the RNA-protein interactions and identify HEV IRESl RNA-binding proteins by screening of the human liver cDNA library using the YBZ1 yeast strain (35).

YBZ1 competent cells were prepared following the lithium acetate method, as previously described (76). To examine self-activation by p3HR2-HEV IRESl, 2 $\mu$g each of the p3HR2-HEV IRESl and pACT2 plasmids or p3HR2 and pACT2 plasmids was cotransformed into the YBZ1 competent cells and plated on LU⁻ medium, as previously described (78). After 3 days of incubation in a humidified incubator at 30°C, eight random colonies were replica plated onto LUH⁻ as well as LUH⁻ plates supplemented with 3-AT (0.1 to 1 mM), followed by 4 days of incubation at 30°C in a humidified incubator. Colonies were monitored on the 4th day. The p3HR2-HEV IRESl-containing colonies showed very little or no growth on LUH⁻ and LUH⁻ + 3-AT medium (Fig. S2 in the supplemental material).

Next, 2 $\mu$g of p3HR2-HEV IRESl plasmid DNA was transformed into YBZ1 competent cells and plated on U⁻ medium. After 3 days of incubation at 30°C, a single colony was inoculated in U⁻ medium and grown at 30°C to prepare the primary culture (YBZ1-p3HR2-HEV IRESl) and subsequently the secondary culture. Aliquots of the culture were used for RNA isolation and verification of HEV-IRESl RNA expression

by RT-qPCR. Next, the secondary culture grown from the primary culture was used for preparation of competent cells of the YBZ1-p3HR2-HEV IRESI.

The human liver Matchmaker cDNA library was procured in the *Escherichia coli* BNN132 strain with a titer of ≥$10^8$ CFU/mL. The cDNA library contained 3.5 × $10^6$ independent clones, with ~93% of colonies containing the cDNA insert. The cDNA library was amplified, and a stock titer of ≥$10^7$ CFU/mL was prepared and stored at −80℃ until further use, following the manufacturer's instructions. Plasmid DNA was isolated from the retitered stock and used for transformation into the YBZ1-p3HR2-HEV IRESI competent cells (the YBZ1 strain containing the p3HR2-HEV IRESI plasmid). A pilot transformation was performed to estimate the efficiency of the cDNA library and to optimize the ideal quantity of DNA required for screening, following the manufacturer's instructions. Approximately, 4 × $10^6$ independent library clones were screened on plates containing LUH⁻ medium supplemented with 0.5 mM 3-AT (0.5 mM 3-AT was added to enhance the stringency of the screening), leading to growth of 395 colonies. Subsequent replica plating of those 395 colonies on LUH⁻ plates containing an increasing concentration of 3-AT (2 mM, 5 mM, 10 mM, and 20 mM) showed that all colonies could grow on 10 mM 3-AT or more. These colonies were inoculated in 5 mL of LU⁻ medium and incubated overnight at 200 rpm and 30℃ in an incubator shaker. The next day, cells were harvested by centrifugation at 7,000 × $g$ and 4℃ for 10 min. Plasmid DNA was isolated using a QIAprep miniprep kit (Qiagen, Hilden, Germany). Briefly, cell pellets were resuspended in buffer P1 (supplied in the kit), followed by 6 freeze-thaw cycles in liquid nitrogen and ambient temperature, respectively. Next, buffer P2 was added, and samples were mixed thoroughly and incubated for 5 min at ambient temperature. Next, buffer N3 was added, and samples were mixed thoroughly and centrifuged at 18,000 × $g$ and 4℃ for 10 min. The supernatant was loaded onto the kit-supplied columns, and plasmid DNA was eluted after a series of washing, as per the manufacturer's instructions. Plasmid DNA isolated from the YBZ1 transformants were transformed into the *E. coli* TOP10 strain, and colonies were grown on LB agar medium containing ampicillin (100 $\mu$g/mL). The presence of the pACT2 plasmid with a cDNA insert sequence in the *E. coli* transformants was verified by colony PCR using the following primers: pACT2 AD-LT seq FP (5′-ATTCGATGATGAAGAT ACCCCA-3′) and pACT2 AD-LT seq RP (5′-GTGAACTTGCGGGGTTTTTCAGTATCTACGA-3′). Two hundred and eighty-five colonies were PCR positive, indicating that they contained a library clone with a cDNA insert. Plasmid DNA was isolated from these colonies and digested with HindIII, XhoI, and EcoRI restriction enzymes to sort them into different categories based on their restriction patterns. Seventy-five clones were found to contain a unique cDNA insert. The insert and its flanking region were sequenced using the following primer: pACT2 AD LT SEQ primer (5′-TGGTGGGGTATCTTCATCATCGAATAG-3′). Analysis of the sequencing data identified 8 unique protein-coding gene sequences in frame with the GAL4 activation domain. These clones were cotransformed along with the p3HR2-HEV IRESI plasmid into YBZ1 competent cells with appropriate controls to ensure that the interaction is reproducible and that there is no self-activation by the prey protein. Liquid $\beta$-galactosidase assays of the colonies were performed as described previously (78).

**RaPID assay.** The RaPID assay was performed as previously described, following similar conditions (34). To identify the HEV IRESI-interacting host proteins, HEK293T cells were cotransfected with the pRMB HEV IRESI and BASU plasmids (in a 6:1 ratio) using Lipofectamine 2000 (1:1 ratio). Forty-two hours posttransfection, 200 $\mu$M biotin-supplemented medium (DMEM + 10% FBS) was added to the cells and maintained for 18 h.

**(i) Cell lysate preparation.** Cells were washed three times with ice-cold phosphate-buffered saline (PBS) and lysed in prechilled radio-immunoprecipitation (RIPA) buffer (150 mM NaCl, 1% NP-40, 0.5% sodium deoxycholate, 0.1% SDS, and 50 mM Tris, pH 8.0) supplemented with protease and phosphatase inhibitor cocktail. The lysate was centrifuged at 14,000 × $g$ for 40 min, and a Macrosep advanced filter (3,000-Da molecule weight cutoff, 20 mL; 89131-974, VWR, USA) was used to remove the free biotin. Clarified supernatant was precipitated in acetone, first at −20℃ for 10 min and then at −80℃ for 20 min. Protein precipitate was solubilized in 8 M urea, and a bicinchoninic (BCA) protein assay kit (Thermo Fisher Scientific, MA, USA) was used to determine the protein concentrations of each sample.

**(ii) Digestion and peptide preparation.** Ten milligrams of protein for each sample was treated with 10 mM dithiothreitol (DTT) for 30 min at 56℃ and alkylated with 20 mM iodoacetamide (IAA) at room temperature for 1 h in the dark. Trypsin (T1426, Thermo Fisher Scientific, MA, USA) was added to the samples at a 1:20 (wt/wt) ratio and incubated at 37℃ for 24 h. Next, 1% formic acid was added to the samples, and desalting of the peptides was done using a Sep-Pak $C_{18}$ cartridge (WAR020515, Waters, MA, USA), followed by lyophilization in a SpeedVac. Lyophilized peptides were solubilized in 1 mL of PBS and incubated with 150 $\mu$L of prewashed streptavidin agarose beads (20361, Thermo Fisher Scientific, MA, USA) for 2 h at ambient temperature. The beads were washed in PBS, followed by washing in wash buffer (5% acetonitrile in PBS) and finally with ultrapure water. Excess liquid was completely removed from the beads, and biotinylated peptides were eluted by adding 0.3 mL of a solution containing 0.1% formic acid and 80% acetonitrile in water by boiling at 95℃ for 5 min. Ten elutions were collected and dried together in a SpeedVac. Enriched peptides were desalted with $C_{18}$ tips (Thermo Fisher Scientific, MA, USA) and reconstituted with solvent A (2% [vol/vol] acetonitrile and 0.1% [vol/vol] formic acid in water) for LC-MS/MS analysis.

**(iii) LC-MS/MS acquisition.** A Sciex 5600⁺ triple time of flight (TOF) mass spectrometer was used for LC-MS/MS analysis. The Sciex 5600⁺ was coupled with a chromXP reversed-phase 3-$\mu$m $C_{18}$-CL trap column (350 $\mu$m × 0.5 mm, 120 Å; Eksigent; AB Sciex, MA, USA) and a nanoViper $C_{18}$ separation column (75 $\mu$m × 250 mm, 3 $\mu$m, 100 Å; Acclaim PepMap; Thermo Fisher Scientific, MA, USA) in an Eksigent nanoLC (ultra 2D plus) system. A binary mobile solvent system was applied with the following composition: solvent C, 2% (vol/vol) acetonitrile and 0.1% (vol/vol) formic acid in water; solvent B, 98% (vol/vol) acetonitrile and 0.1% (vol/vol) formic acid. To separate the peptides, a 60-min gradient was run with a total run time of 90 min at a flow rate of 200 nL/min, and MS data were acquired in information-

dependent acquisition (IDA) with high-sensitivity mode. Each cycle was run for 2.3 s with 250- and 100-ms acquisition times for MS1 ($m/z$ of 350 to 1,250 Da) and MS/MS ($m/z$ of 100 to 1,800) scans, respectively. Every experimental condition was run in quadruplet.

**(iv) Protein identification and quantification.** All raw files (.wiff) were searched in ProteinPilot software (version 4.5; Sciex) with the Mascot algorithm for protein identification and semiquantitation against the Swiss-Prot 57.15 database (20,266 sequences after application of the *Homo sapiens* taxonomy filter). The following parameters were applied to search biotinylated peptides: (i) trypsin as a proteolytic enzyme with a maximum of two missed cleavages allowed; (ii) the allowed tolerance limit for peptide mass error was 20 ppm; (iii) mass error tolerance for fragments was taken up to 0.20 Da; and (iv) carbamido-methylation of cysteine (+ 57.02146 Da), oxidation of methionine (+15.99491 Da), deamination of NQ (+ 0.98416), and biotinylation of lysine (+ 226.07759 Da) were taken as variable modifications. A Pearson correlation plot of peptide intensity was used to monitor data quality between each run and replicate.

**(v) Data analysis.** A protein was considered to be identified if it corresponded to one or more biotinylated peptides, with a posterior error probability (PEP) score of greater than or equal to the median value in the Gaussian smoothing curve of each sample. Note that PEP score refers to the probability that the observed peptide spectrum matches are incorrect. A web-based tool, Bioinformatics and Evolutionary Genomics (http://bioinformatics.psb.ugent.be/webtools/Venn/), was used to identify host proteins that are specific to HEV IRESl RNA and to generate the Venn diagram. The data set with a minimum of 1 unique peptide and a Prot score of 30 or more was considered in the final list of host proteins for further study (Table 1). Prot score refers to the overall protein score generated by Mascot, considering all the observed mass spectra that match amino acid sequences of a certain protein.

**Bioinformatics analysis.** RNA secondary structures were analyzed using the mFold program (http://www.unafold.org/mfold/applications/rna-folding-form.php) based on a minimum free energy calculation at 25°C (79). The host-virus RPPI data set was visualized using Cytoscape (version 3.1.0). The Network Analyzer plug-in in Cytoscape was used to compute the topological parameters and centrality measures of the network. Gene ontology (GO) and Reactome pathway analyses were performed using the Gene Set Enrichment (https://www.gsea-msigdb.org/gsea/downloads.jsp) and Enrichr (https://maayanlab.cloud/Enrichr) (80) tools.

***In silico* analysis of RNA-protein interactions.** The protein structures of the four targets RPS3A, RPS7, RPL5, and RPL26 are not reported; therefore, their sequences were retrieved from the "UniProt" by using their ID numbers P61247, P62081, P46777, and P61254, respectively. Furthermore, the i-tasser module was used to build their three-dimensional (3D) structures. Based on their *c* score, the best model was selected and evaluated through Ramachandran plots. Furthermore, the energy minimization of each protein was performed using the AMBER tool. The 3dRNA tool was used to generate the 3D structure of the RNA sequence. For protein-RNA docking, the HDOCK tool was used with the standard cutoffs. In each docking, the top complex was picked for further analysis. The energy-minimized complex structures were used for quantitative and thermodynamic analysis.

**Luciferase reporter assay, Western blotting analysis, RNA isolation, and real-time quantitative PCR (RT-qPCR).** To perform the dual-luciferase reporter assay, HEK293T cells were seeded at 70% confluence, and the next day, 25 nM siRNA was transfected in respective wells using 0.35 $\mu$L of Dharmafect transfection reagent, as per the manufacturer's instructions (GE Healthcare Dharmacon, Inc., CO, USA). Twelve hours after siRNA transfection, the medium was changed, followed by transfection of the dual-luciferase reporter plasmids (pRLFF/luc87, 1 $\mu$g/well) using Lipofectamine 2000 at a 1:1 (wt/vol) ratio. Eight hours later, the medium was changed, and cells were incubated for 48 h. Cells were washed with ice-cold PBS and used for luciferase assays using the Dual-Glo luciferase assay kit, following the manufacturer's protocol (Promega, WI, USA). The Firefly-Luc values were divided by those of *Renilla*-Luc or divided by the percent cell viability, as specified in the Results, and graphs were plotted in GraphPad Prism or Microsoft Excel. Values are shown as mean ($\pm$standard error of the mean [SEM]) of three independent experiments performed in triplicate. Western blotting was done as previously described (32). Total cellular RNA was isolated using TRI Reagent (MRC, Inc., Cincinnati, OH, USA), as per the manufacturer's protocol. cDNA was synthesized using random hexamers and a commercial cDNA synthesis kit, following the manufacturer's protocol (Firescript cDNA synthesis kit, Solis BioDyne, Tartu, Estonia). RT-qPCR was performed using the SYBR Green detection method using a commercially available kit (5$\times$ Hot Firepol Evagreen qPCR Mix Plus, Solis BioDyne, Tartu, Estonia). The following primers were used to detect the levels of g1-HEV RNA, HEV IRESl RNA, *Renilla*-Luc RNA, and control RNA (nucleotides 618 to 738 in the g1-HEV genome): g1-HEV FP (5'-CGGCCCAGTCTATGTCTCTG-3'), g1-HEV RP (5'-TAGTTCCTGCCTCCCAAAAG-3'), HEV IRESl FP (5'-TATACTCGAGGGTGCCGATCGGTCCC-3'), HEV IRESl RP (5'-TATACCATGGCATCTGGCAGCAAGCTCAG-3'), R-Luc FP (5'-CTTCTTATTTATGGCGACATGTT-3'), R-Luc RP (5'-GCCTGATTTGCCCATACCAATA-3'), control RNA FP (5'-GCCCCCTGGCACATA-3'), and control RNA RP (5'-GAGCGCAGGTTGGAAA-3').

**Ribosome fractionation assay.** Ribosome fractionation was performed as previously described (43, 44). HEK293T cells were seeded at 70% confluence in 100-mm dishes and transfected with pGL3-control RNA, pSuper-HEV IRESl, and pRL-TK plasmid DNA (12 $\mu$g/dish) using Lipofectamine 2000 (1:1 ratio). Forty-eight hours posttransfection, 100 $\mu$g/mL cycloheximide (CHX) or 0.5 $\mu$g/mL puromycin (Puro) was added to the culture medium, and incubation was continued for 30 min at 37°C with 5% $CO_2$. Next, cells were washed three times with ice-cold PBS (containing 100 $\mu$g/mL CHX for CHX treatment), followed by trypsinization and centrifugation for 5 min at 500 $\times$ *g*. The supernatant was discarded, and the cell pellet was resuspended in 1 mL of polysome extraction buffer (PEB; 20 mM Tris-HCl [pH 7.5], 100 mM KCl, 5 mM $MgCl_2$, 0.5% NP-40, 100 $\mu$g/mL CHX, 1$\times$ protease inhibitors, and a 1:1,000 dilution of RiboLock RNase inhibitor), incubated on ice for 30 min with occasional mixing, and centrifuged at 12,000 $\times$ *g* for 20 min at 4°C. Total protein and RNA concentration in the clarified lysate were measured by Bradford assay or Nanodrop

spectrophotometry, respectively. Aliquots of the lysate were used for cDNA synthesis, Western blotting, and ribosome fractionation.

A linear 10 to 50% sucrose density gradient was prepared for ribosome fractionation 1 day before the polysome fractionation experiment. Stock solutions of 10%, 20%, 30%, 40%, and 50% sucrose were prepared using 2.2 M sucrose and 10× salt solution (1000 mM NaCl, 200 mM Tris-HCl [pH 7.5], and 50 mM $MgCl_2$) in diethyl pyrocarbonate (DEPC)-treated water. Then, 2.2 mL of 10% stock solution of sucrose was added to the bottom of a thin-walled, 12.5-mL ultracentrifuge tube, followed by the addition of 2.2 mL each of 20%, 30%, 40%, and 50% sucrose solutions. Tubes were stored at 4°C overnight for linearization. The next day, equal amounts of clarified lysate were loaded to the top of the tubes, and weights were equalized. Tubes were centrifuged in a SW40Ti swinging bucket rotor at 190,000 × $g$ for 90 min at 4°C using a Beckman ultracentrifuge (Beckman Coulter, CA, USA). After centrifugation, tubes were set for fractionation in the automated fraction collection unit, and 24 fractions of 500 $\mu$L each were collected. Profiles of RNA in the fractions were captured as an indicator of different ribosome fractions (shown in Fig. S4A). Alternate fractions (1 to 23) were selected for further analysis of RNA and protein distribution profiles.

**$m^7$G-capped RNA and biotinylated RNA pulldown assay.** For $m^7$G-capped RNA pulldown experiments, HEK293T cells were seeded in 60-mm dishes at a confluence of 70%. The next day, cells were cotransfected with 2 $\mu$g each of the pSuper-HEV IRESI and pRL-TK plasmids, and cells were incubated for 48 h in DMEM supplemented with 10% FBS in a 37°C incubator supplied with 5% $CO_2$. Next, cells were lysed in 1 mL of TRI Reagent, and RNA was extracted as per the manufacturer's instructions (MRC, Inc., Cincinnati, OH, USA). Twenty micrograms of total RNA was added to the binding buffer (1% Triton X-100, 150 mM NaCl, 2 mM EDTA [pH 8.0], and 20 mM Tris-HCl [pH 8.0]), 20 $\mu$L of monoclonal anti-$m^7$G cap antibody (Merck, MA, USA), and 1 $\mu$L of RNase inhibitor (RNAsin, Promega, WI, USA) was added, and samples were incubated overnight at 4°C with continuous mixing. Twenty-five microliters of Protein G Plus agarose beads (Santa Cruz Biotechnology, Dallas, TX, USA) were washed twice in binding buffer and added to the samples, followed by a 1-h incubation. Samples were centrifuged at 1,000 rpm for 1 min at 4°C. Supernatant containing unbound RNA was collected, and RNA was extracted using TRI Reagent and stored at −80°C. Beads were washed four times in ice-cold low-salt wash buffer (1% SDS, 1% Triton X-100, 150 mM NaCl, 2 mM EDTA [pH 8.0], and 20 mM Tris-HCl [pH 8.0]), once in high-salt wash buffer (1% SDS, 1% Triton X-100, 500 mM NaCl, 2 mM EDTA [pH 8.0], and 20 mM Tris-HCl [pH 8.0]), and twice with Tris-EDTA buffer (pH 8.0). Finally, beads were resuspended in elution buffer (1% SDS and 0.1 M $NaHCO_3$), incubated for 5 min at 4°C on a rocker, and centrifuged at 1,000 rpm for 1 min at 4°C, and the supernatant was collected. RNA was isolated from the supernatant using TRI Reagent, and all samples were simultaneously processed for RT-qPCR.

For biotinylated RNA pulldown assays, $10^7$ Huh7 cells were lysed in binding buffer (1% Triton X-100, 0.5% NP-40, 150 mM NaCl, 2 mM EDTA [pH 8.0], 20 mM Tris-HCl [pH 8.0], 1× complete protease inhibitor cocktail, and 40 U RNase inhibitor), and clarified lysates were incubated with biotinylated control RNA (control RNA [B$^+$]), nonbiotinylated HEV IRESI RNA (HEV IRESI RNA [B$^-$]), biotinylated HEV IRESI RNA (HEV IRESI RNA [B$^+$]), or mock lysate (cell lysate without RNA) for 1 h on a flip-flop rocker at 4°C. Streptavidin magnetic beads were washed twice in binding buffer and added to the RNA-protein mixture, followed by incubation for 1 h, under similar conditions. Streptavidin-bound RNA-protein complexes were washed two times in ice-cold low-salt wash buffer (1% SDS, 1% Triton X-100, 150 mM NaCl, 2 mM EDTA [pH 8.0], and 20 mM Tris-HCl [pH 8.0]), once in high-salt wash buffer (1% SDS, 1% Triton X-100, 250 mM NaCl, 2 mM EDTA [pH 8.0], and 20 mM Tris-HCl [pH 8.0]), and once with Tris-EDTA buffer (pH 8.0). Fifty percent of the beads were resuspended in elution buffer (1% SDS and 0.1 M $NaHCO_3$) and incubated for 5 min at 4°C on a rocker, followed by collection of the supernatant for checking the presence of RNA by formaldehyde-agarose gel electrophoresis. The rest of the beads were mixed with 2× Laemmli buffer and incubated at 95°C for 5 min, followed by collection of the supernatant for Western blotting using various antibodies.

**Statistical analysis.** Data are shown as mean ± standard error of mean (SEM) of three independent experiments. $P$ values were calculated by two-tailed Student's $t$ tests (paired two samples for means).

**Data availability.** The mass spectrometry proteomics data have been deposited to the ProteomeXchange Consortium via the PRIDE partner repository with the data set identifier PXD031009.

## SUPPLEMENTAL MATERIAL

Supplemental material is available online only.
**SUPPLEMENTAL FILE 1**, PDF file, 1.4 MB.

## ACKNOWLEDGMENTS

The RNA motif plasmid cloning backbone (pRMB) and BASU RaPID plasmid were kindly gifted by Paul Khavari. We thank Marvin Wickens for providing the Y3H assay system. We thank Charles M. Rice for providing the g3A HCV replicon. We thank Dipanka Tanu Sarmah and Samrat Chatterjee for data analysis using Enrichr. We thank Chandru Subramani for helping with the initial experiments. We appreciate continuous support by the technical staff of the Regional Centre for Biotechnology mass spectrometry facility for data collection.

This study was funded by the Science and Engineering Research Board (SERB), Government of India, grant to M.S. and a THSTI core grant to M.S. S.K. is supported by a senior research fellowship from the Department of Biotechnology, Government of India. R.V. and S.S. are supported by a senior research fellowship from the Council of Scientific and Industrial Research, Government of India.

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
