## [Reviewer comments · Microbiology Spectrum]

Microbiology Spectrum

RNA-protein interactome at the Hepatitis E virus internal ribosome entry site

Shiv Kumar, Rohit Verma, Sandhini Saha, Ashish Agrahari, Shivangi Shukla, Oinam Singh, Umang Berry, . Anurag, Tushar Maiti, Shailendra Asthana, C.T. Ranjith-Kumar, and Milan Surjit

Corresponding Author(s): Milan Surjit, Translational Health Science and Technology Institute

Review Timeline:

Submission Date:	July 22, 2022
Editorial Decision:	September 16, 2022
Revision Received:	December 15, 2022
Editorial Decision:	January 13, 2023
Revision Received:	January 16, 2023
Editorial Decision:	April 18, 2023
Revision Received:	April 18, 2023
Accepted:	June 5, 2023

Editor: Saumitra Das

Reviewer(s): Disclosure of reviewer identity is with reference to reviewer comments included in decision letter(s). The following individuals involved in review of your submission have agreed to reveal their identity: Vasudevan Sheshadri (Reviewer #2)

Transaction Report:

DOI: <https://doi.org/10.1128/spectrum.02827-22>

September 16, 2022

Prof. Milan Surjit
Translational Health Science and Technology Institute
Faridabad
India

Re: Spectrum02827-22 (**RNA-protein interactome at the Hepatitis E virus internal ribosome entry site**)

Dear Prof. Surjit,

Thank you for submitting your manuscript to Microbiology Spectrum.

While both reviewers have found the work interesting, there are several concerns that need to be addressed. I would like you to revise the manuscript in line with the reviewers' comments.

Link Not Available

Sincerely,

Saumitra Das

Journals Department
Reviewer comments:

Reviewer #1 (Comments for the Author):

The study is aimed to identify the proteins that can bind IRES of Hepatitis E virus. The authors also investigate the role of these proteins in IRES activity. They have used RaPID and yeast 3-hybrid assays for this purpose. The experimental design and approach they have used for the identification is very robust and convincing. That is the best part of this study. However, the study fails to convincingly demonstrate the role of these IRES-interacting proteins in IRES activity. Thus, their conclusions on the functional part of these proteins are not supported by their results.

Major comments:

1. Fig 6 D and E: Results show that RPL5 is required for both cap-dependent and cap-independent translation. Fig 8 A and B: Results show that RPL5 is important for the association of both capped and uncapped (IRES) mRNAs to with polysomes (translating pool). These results clearly indicate that global translation (capped and uncapped mechanisms) is affected when RPL5 is knocked down, which is expected as RPL5 is an important ribosome protein. Thus, RPL5 does not seem to have a IRES-specific role based on the experiments shown in this study. The observation can be explained by reduced overall translation level due to a lack of an important ribosomal protein. In other words, importance of interaction of RPL5 with HEV IRES remains to be understood.
2. Fig 6 D and E: Results show that DHX9 is important for cap-dependent translation (RLuc). Fig 8 B: Results show that DHX9 is not important for the association of capped (RLuc) mRNAs to associate with polysomes (translating pool). These two results are not consistent with each other, rather they seem to contradict each other.
3. I strongly recommend the authors to explore other identified proteins for their potential role in IRES activity. The proteins they have characterized in this manuscript have not shown any specific and direct role in IRES-mediated translation.
4. Fig 6F: These results can also be explained by reduced overall translation level expected due to lack of important ribosome proteins and the helicase.
5. Fig. 8: Show polysome profile in knockdown cells.
6. What is the control RNA used in assays described in Figure 4 and 7D? Fig. 4C: A negative control (i.e., a protein not predicted to bind the IRES) would be required.
7. Figure 5I: A negative control (i.e., a protein not predicted to bind the IRES) will help to appreciate the binding affinity.

Minor comments:

1. Font size in all Figures is too small to read clearly.
2. Line 350: "Above network parameters suggest that the network is highly connected". Please explain the logic/reasoning behind this.
3. Expand PPI

Reviewer #2 (Comments for the Author):

Review of the article titled "RNA-protein interactome at the Hepatitis E virus internal ribosome entry site" by Kumar et al describe the identification of host proteins that interacts with the previously identified IRES like element of g1-HEV. The authors took two approaches to identify the interacting proteins RAPID method that identifies the RNA interacting proteins by virtue of the Bir ligase docked in the vicinity and the yeast three hybrid approaches. The further characterized some of these identified host proteins and showed that depletion of two of these proteins results in reduced viral replication. While the results are clearly presented and are supported by the data I have the following minor concerns/comments.

1. One major concern is lack of any data to support that any of the identified protein is directly interacting with the RNA, as many of the identified proteins are components of the larger complex. Even in case of yeast 3 hybrid it is possible that the identified partners are part of the larger ribosome complex from yeast and may not be directly interacting with the IRESI element. RNA immunoprecipitation following crosslinking with the identified proteins would have revealed which of the proteins are directly in contact with the IRESI RNA.
2. Another concern is except for one protein (RPL26) there is no overlap between the candidates identified by the two approaches. Can the authors explain the possible reasons for the same.
3. Another point is in Fig 5 it shows that siRNA for DHX9 reduces the translation of both cap dependent (Renila) and independent (Luc) translation while in the polysome profiling assay it affects the association of only the cap independent RNA and not cap dependent RNA (Fig 8 A and B).
4. A minor point is as per the reporter assay (Fig5), not of the interacting proteins specifically affected only the CAP independent translation, the three proteins RPL5, 26 and 9 all reduced both the cap dependent and independent translation suggesting that these may play a role in translation and not necessarily only in cap IRESI mediated translation.
5. since many of the interacting proteins were ribosomal, it is possible that the interacting proteins are part of the ribosome when interacting with IRESI. Did the authors check for the presence of non interacting ribosomal proteins in the biotin pull down (Fig 4C) or when assessing the polysome association (Fig 7 and 8)
6. a minor comment is that the authors state that they identified 43 unique proteins by RAPID assay, but it is difficult to understand this point from the venn diagram presented in Fig2. The figure legends need to be slightly more elaborate to clearly understand.
7. Lastly out of curiosity did the authors perform the Viral replication assay or the reporter translation assay in the presence of ER stress and specific siRNA depletion

Staff Comments:

Preparing Revision Guidelines

Please return the manuscript within 60 days; if you cannot complete the modification within this time period, please contact me. If you do not wish to modify the manuscript and prefer to submit it to another journal, please notify me of your decision immediately so that the manuscript may be formally withdrawn from consideration by Microbiology Spectrum.

Thanks again for submitting your paper to Microbiology Spectrum.

Point wise response to Reviewer comments

Reviewer #1 (Comments for the Author):

The study is aimed to identify the proteins that can bind IRES of Hepatitis E virus. The authors also investigate the role of these proteins in IRES activity. They have used RaPID and yeast 3-hybrid assays for this purpose. The experimental design and approach they have used for the identification is very robust and convincing. That is the best part of this study. However, the study fails to convincingly demonstrate the role of these IRES-interacting proteins in IRES activity. Thus, their conclusions on the functional part of these proteins are not supported by their results.

Response: We thank the reviewer for the constructive comments. We have performed additional experiments, added new data and revised the manuscript text and figures to improve the overall conclusion.

Major comments:

1. Fig 6 D and E: Results show that RPL5 is required for both cap-dependent and cap-independent translation. Fig 8 A and B: Results show that RPL5 is important for the association of both capped and uncapped (IRES) mRNAs to with polysomes (translating pool). These results clearly indicate that global translation (capped and uncapped mechanisms) is affected when RPL5 is knocked down, which is expected as RPL5 is an important ribosome protein. Thus, RPL5 does not seem to have a IRES-specific role based on the experiments shown in this study. The observation can be explained by reduced overall translation level due to a lack of an important ribosomal protein. In other words, importance of interaction of RPL5 with HEV IRES remains to be understood.

Response: We agree with reviewer that our results do not suggest HEV IRES-specific role of RPL5. Being an essential ribosomal protein, it is perhaps required for most modes of translation. Therefore, its association with a putative internal ribosome entry-site (HEV-IRESI) supports the ability of the HEV-IRESI as a genuine IRES.

We would like to note that our previous publication (Nair et al., PLOS Pathogens, 2016) on identification of the HEV-IRESI did not demonstrate that it was a genuine IRES element. We did not find close similarity of HEV-IRESI with any of the wellknown IRES sequence, except for weak homology with ERAV-IRES (Nair et al., PLOS Pathogens, 2016). We had done preliminary characterization studies, which demonstrated that HEV-IRESI was capable of initiating cap-independent translation. Therefore, we had named it as IRES-like element.

The main objective of this manuscript was to establish whether HEV IRESI is a genuine IRES element. Its association with multiple ribosomal proteins as well as its presence in the polysome fraction supports that HEV IRESI is indeed a genuine IRES, which drives cap-independent translation. siRNA-treatment mediated depletion of ribosomal proteins reduces the level of polysomes and association of HEV-IRES with polysomes, further supporting the function of HEV-IRES as a genuine IRES. We used Renilla-luc as a marker of cap-dependent translation to show that those ribosomal proteins are important for translation.

In the revised manuscript, we have also added the data for HCV-IRES (which is a well-characterized IRES element) as an additional control. It was observed that lack of RPL5 did not

reduce HCV-IRES dependent translation (Fig. 7D). Thus, although RPL5 is not specific to HEV-IRES function, it is not essential for the function of all IRES elements.

2. Fig 6 D and E: Results show that DHX9 is important for cap-dependent translation (RLuc). Fig 8 B: Results show that DHX9 is not important for the association of capped (RLuc) mRNAs to associate with polysomes (translating pool). These two results are not consistent with each other, rather they seem to contradict each other.

Response: Based on the data, it appears that DHX9 controls cap-dependent translation independent of the association of capped RNAs with polysome. It might be inhibiting protein synthesis by interfering at a step downstream of polysome assembly.

3. I strongly recommend the authors to explore other identified proteins for their potential role in IRES activity. The proteins they have characterized in this manuscript have not shown any specific and direct role in IRES-mediated translation.

Response: We have added the data on one more interaction partner of HEV-IRES, which is ribosomal protein S7 (RPS7). Lack of RPS7 does not affect cell viability or global translation (Fig. 6). Lack of RPS7 does not affect cap-dependent translation or HEV-IRES dependent translation (Fig. 7B and C). However, lack of RPS7 significantly reduces HCV-IRES dependent translation (Fig. 7D).

Collectively, these data suggest that different set of ribosomal subunit proteins control translation initiated through different mechanisms. Our data also show that although RPL5 and DHX9 are not specific only to HEV-IRES mediated translation, they are essential for HEV-IRES mediated translation.

We are exploring the role of additional interaction partners of HEV-IRES, with the hope to identify a HEV-IRES specific host factor.

4. Fig 6F: These results can also be explained by reduced overall translation level expected due to lack of important ribosome proteins and the helicase.

Response: Following two experiments were performed to determine whether the observed inhibitory effect of RPL-5 and DHX9 siRNAs on HEV replication was due to a reduced overall translation. (a) We measured the level of global translation in siRNA transfected cells by transient L-Azidohomoalanine labelling. There is around 25-30% reduction in global translation in RPL-5 and DHX9 siRNA transfected cells (Fig. 6C). However, there is 60% reduction in HEV replication in RPL-5 and DHX9 siRNA transfected cells (Fig. 7E). We also measured the activity of HCV IRES and there was no significant change in its activity in RPL-5 and DHX9 siRNA transfected cells (Fig. 7D). In contrast, there was significant reduction in HCV-IRES activity in RPL41 and RPS7 siRNA treated cells (Fig. 7D). These data suggest that global translation was not shut down in siRNA transfected cells and the ribosomal proteins analysed in our study does not control all modes of translation.

5. Fig. 8: Show polysome profile in knockdown cells.

Response: We regret to inform that we could not retrieve the polysome profiles from the computer connected with the equipment due to accidental breakdown of the entire system. It will take long time to get it repaired.

6. What is the control RNA used in assays described in Figure 4 and 7D? Fig. 4C: A negative control (i.e., a protein not predicted to bind the IRES) would be required.

Response: Sequence from 618-738 nucleotides in the g1-HEV genome was used as control RNA (described in the manuscript text and methods). GAPDH was used as a negative control to check specificity of the pull down assay (Fig. 4C).

7. Figure 5I: A negative control (i.e., a protein not predicted to bind the IRES) will help to appreciate the binding affinity.

Response: GAPDH was used as a control protein to appreciate the binding affinity. Values obtained for GAPDH was used as cut off to rule out non specific signal (Fig. 5 E,J,K)

Minor comments:

1. Font size in all Figures is too small to read clearly.

Response: Font size has been increased in the revised manuscript figures.

2. Line 350: "Above network parameters suggest that the network is highly connected". Please explain the logic/reasoning behind this.

Response: In order to predict the structure of multiprotein complexes, protein-protein interaction (PPI) datasets are analysed using computational approaches to build the PPI network. More the connectivity among the interaction partners of a PPI network, denser is the network (Degree). PPI network is indicated by the no. of edges (interactions) with reference to a given no. of nodes (proteins). Proteins that share a large number of interactions are represented as a cluster. Thus, clusters show the structure of the PPI network and analysis of the clusters suggest possible functions of the protein complex present in the network. Therefore, measurement of degree, edges and clustering coefficient reflect the connectivity and possible importance of the PPI network.

3. Expand PPI

Response: PPI: protein-protein interaction. We have expanded it in both figure legend and results sections in the revised manuscript.

Reviewer #2 (Comments for the Author):

Review of the article titled "RNA-protein interactome at the Hepatitis E virus internal ribosome entry site" by Kumar et al describe the identification of host proteins that interacts with the previously identified IRES like element of g1-HEV. The authors took two approaches to identify the interacting proteins RAPID method that identifies the RNA interacting proteins by virtue of the

Bir ligase docked in the vicinity and the yeast three hybrid approaches. The further characterized some of these identified host proteins and showed that depletion of two of these proteins results in reduced viral replication. While the results are clearly presented and are supported by the data I have the following minor concerns/comments.

Response: We thank the reviewer for the constructive comments. We have performed additional experiments, added new data and revised the manuscript text and figures to improve the overall conclusion.

1. One major concern is lack of any data to support that any of the identified protein is directly interacting with the RNA, as many of the identified proteins are components of the larger complex. Even in case of yeast 3 hybrid it is possible that the identified partners are part of the larger ribosome complex from yeast and may not be directly interacting with the IRES element. RNA immunoprecipitation following crosslinking with the identified proteins would have revealed which of the proteins are directly in contact with the IRES RNA.

Response: We agree with the reviewer that RNA immunoprecipitation following crosslinking is a better technique to identify direct interaction partners of the HEV IRES RNA. Perhaps we could not optimize the assay in our laboratory. Western blots were not very clear after crosslinking. So we did not pursue that experiment.

2. Another concern is except for one protein (RPL26) there is no overlap between the candidates identified by the two approaches. Can the authors explain the possible reasons for the same.

Response: We believe that there are multiple reasons behind little overlap between the candidates identified by the two approaches: Y3H generally identifies one direct interaction partner in one yeast cotransformant and RaPID identifies a protein complex consisting of RNA binding proteins as well as secondary protein interaction partners of those RNA binding proteins. RaPID assay depends on biotinylation of lysines exposed on the surface of RNA-binding proteins present within 10 nm distance from the BirA ligase. Thus, it is possible to lose some of the RNA binding proteins if they are buried deep inside the complex and inaccessible to BirA ligase for biotinylation. Quality of the cDNA library also influences the end result of a Y3H library screening. Overrepresentation of some RNA binding proteins in the cDNA library may mask the identification of other potential interaction partners. Similarly, quality of the LC-MS data influences the end result of RaPID assay.

3. Another point is in Fig 5 it shows that siRNA for DHX9 reduces the translation of both cap dependent (Renila) and independent (Luc) translation while in the polysome profiling assay it affects the association of only the cap independent RNA and not cap dependent RNA (Fig 8 A and B).

Response: We do not have a direct answer to explain this observation. It is possible that DHX9 serves distinct functions during HEV IRES-dependent and cap-dependent translation. We have added new data on the effect of DHX9 siRNA on HCV IRES activity, which shows its dispensability during HCV IRES-mediated translation (Fig. 7D). Additional studies are required to understand the different modes of action of DHX9.

4. A minor point is as per the reporter assay (Fig5), not of the interacting proteins specifically affected only the CAP independent translation, the three proteins RPL5, 26 and 9 all reduced

both the cap dependent and independent translation suggesting that these may play a role in translation and not necessarily only in cap IRESI mediated translation.

Response: We agree with reviewer that our results do not suggest HEV IRES-specific role of RPL5, RPL26 and DHX9. Being essential ribosomal proteins and RNA helicase, these proteins are likely to be required for translation. Therefore, their association with a putative internal ribosome entry-site (HEV IRESI) supports the claim that the HEV IRESI is a genuine IRES element. We would like to note that our previous publication (Nair et al., PLOS Pathogens, 2016) on identification of the HEV IRESI did not demonstrate that it was a genuine IRES element. We did not find close similarity of HEV IRESI with any of the wellknown IRES sequence, except for weak homology with ERAV IRES (Nair et al., PLOS Pathogens, 2016). We had done preliminary characterization studies, which demonstrated that HEV IRESI was capable of initiating cap-independent translation. Therefore, we had named it as IRES-like element.

The main objective of this manuscript was to establish whether HEV IRESI is a genuine IRES element. Its association with multiple ribosomal proteins as well as its presence in the polysome fraction supports that HEV IRESI is indeed a genuine IRES, which drives cap-independent translation. siRNA-treatment mediated depletion of ribosomal proteins reduces the level of polysomes and association of HEV IRES with polysomes, further supporting the function of HEV IRES as a genuine IRES. We used Renilla-luc as a marker of cap-dependent translation to show that those ribosomal proteins are important for translation.

In the revised manuscript, we have also added the data for HCV IRES (which is a well-characterized IRES element) as an additional control. It was observed that lack of RPL5, RPL26 or DHX9 did not reduce HEV IRES-dependent translation (Fig. 7D). Thus, although RPL5, RPL26 or DHX9 are not specific to HEV IRES function, they are not essential for the function of all IRES elements.

5. since many of the interacting proteins were ribosomal, it is possible that the interacting proteins are part of the ribosome when interacting with IRESI. Did the authors check for the presence of non interacting ribosomal proteins in the biotin pull down (Fig 4C) or when assessing the polysome association (Fig 7 and 8)?

Response: We have not checked the presence of non interacting ribosomal proteins in the biotin pull down assay.

6. a minor comment is that the authors state that they identified 43 unique proteins by RAPID assay, but it is difficult to understand this point from the venn diagram presented in Fig2. The figure legends need to be slightly more elaborate to clearly understand.

Response: We have tried to improve the legend as suggested by the reviewer.

7. Lastly out of curiosity did the authors perform the Viral replication assay or the reporter translation assay in the presence of ER stress and specific siRNA depletion?

Response: We have not measured viral replication or IRES reporter activity in the presence of ER stress and specific siRNA depletion.

January 13, 2023

Prof. Milan Surjit
Translational Health Science and Technology Institute
Faridabad
India

Re: Spectrum02827-22R1 (**RNA-protein interactome at the Hepatitis E virus internal ribosome entry site**)

Dear Prof. Milan Surjit:

Both the reviewers have recommended acceptance of the manuscript. However, Reviewer 2 has expressed some concerns, which need to be addressed before final acceptance.

Particularly, the authors should restrict the claims to only IRES binding and a role in translation in general (Both cap-dependent and HEV-IRES). Also, the explanation regarding the discrepancy (Fig 6 and 8) should be more convincing as it is not clear how translation could be affected without affecting the polysome association of mRNA.

These issues can be clarified appropriately in the discussion section.

Link Not Available

Sincerely,

Saumitra Das

Journals Department
Reviewer comments:

Reviewer #2 (Comments for the Author):

One major concern is still regarding the specificity of the identified proteins role in HEV-IRES dependent translation as all the assays indicate that the identified interacting proteins affect both cap dependent and HEV-IRES dependent translation. Further the new data regarding the RPS7 (or RPL5 and HCV-IRES translation assay) actually reinforces this observation as it does not affect cap dependent and HEV-IRES translation but affects the HCV-IRES translation and Vice versa. However the interaction of these proteins with the HEV IRES seems to be specific and the data for the same is convincing.

Staff Comments:

Preparing Revision Guidelines

Please return the manuscript within 60 days; if you cannot complete the modification within this time period, please contact me. If you do not wish to modify the manuscript and prefer to submit it to another journal, please notify me of your decision immediately so that the manuscript may be formally withdrawn from consideration by Microbiology Spectrum.

Point wise response to Reviewer comments

Reviewer #2 (Comments for the Author):

One major concern is still regarding the specificity of the identified proteins role in HEV-IRES dependent translation as all the assays indicate that the identified interacting proteins affect both cap dependent and HEV-IRES dependent translation. Further the new data regarding the RPS7 (or RPL5 and HCV-IRES translation assay) actually reinforces this observation as it does not affect cap dependent and HEV-IRES translation but affects the HCV-IRES translation and Vice versa. However the interaction of these proteins with the HEV IRES seems to be specific and the data for the same is convincing.

Response:

We thank the reviewer for approving the revised manuscript.

Based on the comment of the reviewer, we have modified the text (lines 36-39; 610-624; 676-677) and added one reference (63). We hope that the current manuscript clarifies the remaining concerns of the reviewer and further improves the manuscript.

April 18, 2023

Prof. Milan Surjit
Translational Health Science and Technology Institute
Faridabad
India

Re: Spectrum02827-22R2 (**RNA-protein interactome at the Hepatitis E virus internal ribosome entry site**)

Dear Prof. Milan Surjit:

Thank you for submitting your corrected manuscript to Microbiology Spectrum.
I have gone through the corrected version.

My observations are as follows:

Authors have changed the figure 3 R9, R10, R11 and R26 which represents the B-gal activity of Yeast Hybrid assay system. The authors also mentioned that they used the same colonies for culture in the liquid media and measured the B-Gal activity which has been plotted parallelly. The earlier figures showed growth in R10, R11 at 5mM 3AT but the new figures show very little growth, but the relative B-Gal activity seems to remain the same upon using the same colonies in liquid media in the new figure and also in the old figure.

I have also contacted one of the reviewer and the comments are as follows:

Reviewer 1:

I cross checked the Fig 3 with the text for consistency in explanation. The growth assays (colonies) are consistent with their explanation in the text. But, the galactosidase assay is not.
For example, compare R10 and R7 (and few other places). I am confused about their interpretation.

Link Not Available

Sincerely,

Saumitra Das

Journals Department
Reviewer comments:

Reviewer #1 (Comments for the Author):

Authors have responded satisfactorily to all the queries

Staff Comments:

Preparing Revision Guidelines

Please return the manuscript within 60 days; if you cannot complete the modification within this time period, please contact me. If you do not wish to modify the manuscript and prefer to submit it to another journal, please notify me of your decision immediately so that the manuscript may be formally withdrawn from consideration by Microbiology Spectrum.

Point wise response to the comments of the editor and reviewers

My observations are as follows:

Authors have changed the figure 3 R9, R10, R11 and R26 which represents the B-gal activity of Yeast Hybrid assay system. The authors also mentioned that they used the same colonies for culture in the liquid media and measured the B-Gal activity which has been plotted parallelly. The earlier figures showed growth in R10, R11 at 5mM 3AT but the new figures show very little growth, but the relative B-Gal activity seems to remain the same upon using the same colonies in liquid media in the new figure and also in the old figure.

Author response: In the revised figure 3, we changed the image in row 9, column 3, row 10, columns 2 and 3; row 11, columns 2 and 3; row 26, columns 2 and 3. We agree that in rows 10 and 11, there is less colony growth in column 3, compared to the old figure. As shown earlier, HCV-IRES+PPIG shows more growth than HEV-IRESI+PPIG.

We have not changed the beta galactosidase assay graph because the graph was generated using colonies from the current figure (revised one). Older image contained wrong panels which was inserted by mistake during preparation of the figure initially. So both graph and colony growth data is generated from same colonies.

We agree that there is discrepancy between growth of colony in 3-AT containing medium and beta-galactosidase reporter data. Beta-galactosidase data was reproducible and we did not observe much difference between. However, this discrepancy is not limited to only rows 10 and 11, neither attributed to change of the images in the revised figure. For example, in row 23 and row 31, even though the colony growth is robust at 10mM 3AT, beta galactosidase activity is less than that seen in row 18.

We would like to clarify that in the Yeast three hybrid assay, his3 and beta-galactosidase reporters are independently controlled and present on different genomic loci. That is useful to rule out potential biasness in the assay and helps in providing two independent readouts of the interaction. However, depending on the nature of the interaction partners being studied, variations in read out is possible due to the influence of adjacent elements in the genome. Therefore, even though beta-galactosidase readings are quantitative, they do not always quantitatively reflect upon the strength of interaction. Here, we relied upon growth of colonies on 3-AT medium to measure the strength of interaction because in that case growth pattern could be followed in a dose dependent manner. We have added this explanation in the manuscript text to clarify the interpretation.

I have also contacted one of the reviewer and the comments are as follows:

Reviewer 1:

I cross checked the Fig 3 with the text for consistency in explanation. The growth assays (colonies) are consistent with their explanation in the text. But, the galactosidase assay is not.

For example, compare R10 and R7 (and few other places). I am confused about

their interpretation.

Author response: We agree that there is discrepancy between growth of colony in 3-AT containing medium and beta-galactosidase reporter data. However, this discrepancy is not limited to only rows 10 and 11, neither attributed to change of the images in the revised figure. For example, in row 23 and row 31, even though the colony growth is robust at 10mM 3AT, beta galactosidase activity is less than that seen in row 18.

We would like to clarify that in the Yeast three hybrid assay, his3 and beta-galactosidase reporters are independently controlled and present on different genomic loci. That is useful to rule out potential biasness in the assay and helps in providing two independent readouts of the interaction. However, depending on the nature of the interaction partners being studied, variations in read out is possible due to the influence of adjacent elements in the genome. Therefore, even though beta-galactosidase readings are quantitative, they do not always quantitatively reflect upon the strength of interaction. Here, we relied upon growth of colonies on 3-AT medium to measure the strength of interaction because in that case growth pattern could be followed in a dose dependent manner. We have added this explanation in the manuscript text to clarify the interpretation.

June 5, 2023

Prof. Milan Surjit
Translational Health Science and Technology Institute
Faridabad
India

Re: Spectrum02827-22R3 (**RNA-protein interactome at the Hepatitis E virus internal ribosome entry site**)

Dear Prof. Milan Surjit:

Your manuscript has been accepted, and I am forwarding it to the ASM Journals Department for publication. You will be notified when your proofs are ready to be viewed.

As the reviewer 1 has suggested, please provide detailed explanation for the observations made in Figure 3 in the manuscript also. The explanation is not just for Reviewers.
This minor modification can be incorporated in the final version.

Sincerely,

Saumitra Das
Editor, Microbiology Spectrum
